# Discovering Influential Neuron Path in Vision Transformers

**Yifan Wang[1], Yifei Liu[1], Yingdong Shi[1], Changming Li[1],**
**Anqi Pang[2], Sibei Yang[1], Jingyi Yu[1], Kan Ren**[*1]
[1]ShanghaiTech University, [2]Tencent PCG
{wangyf7,renkan}@shanghaitech.edu.cn

## Abstract

Vision Transformer models exhibit immense power yet remain opaque to human understanding, posing challenges and risks for practical applications. While prior research has attempted to demystify these models through input attribution and neuron role analysis, there's been a notable gap in considering layer-level information and the holistic path of information flow across layers. In this paper, we investigate the significance of influential neuron paths within vision Transformers, which is a path of neurons from the model input to output that impacts the model inference most significantly. We first propose a joint influence measure to assess the contribution of a set of neurons to the model outcome. And we further provide a layer-progressive neuron locating approach that efficiently selects the most influential neuron at each layer trying to discover the crucial neuron path from input to output within the target model. Our experiments demonstrate the superiority of our method finding the most influential neuron path along which the information flows, over the existing baseline solutions. Additionally, the neuron paths have illustrated that vision Transformers exhibit some specific inner working mechanism for processing the visual information within the same image category. We further analyze the key effects of these neurons on the image classification task, showcasing that the found neuron paths have already preserved the model capability on downstream tasks, which may also shed some lights on real-world applications like model pruning. The project website including implementation code is available at https://foundation-model-research.github.io/NeuronPath/.

## 1 Introduction

Transformer (Vaswani et al., 2017) models in the vision domain, such as supervised Vision Transformers (Dosovitskiy et al., 2021) (ViT) or self-supervised pretrained models (He et al., 2022; Oquab et al., 2023), have showcased remarkable performance in various real-world tasks like image classification (Dosovitskiy et al., 2021) and image synthesis (Peebles & Xie, 2023). However, the inner workings of these vision Transformer models remain elusive, despite their impressive achievements. Understanding the internal mechanisms of vision models is crucial for both research and practical applications. When confronted with the model decision outputs, one may raise some questions that, *how is the vision Transformer model processing the input information by layer, and which part of the model is significant to derive the final outcome*? Unraveling the synergy within these models is essential for comprehending machine learning systems. The current lack of complete understanding poses challenges and risks when deploying these models in real-world scenarios.

Literature has explored mechanism discovery and explainability on vision models. Previous works about the explainability of vision models mostly focused on the visualization of inner patterns of models (Zeiler & Fergus, 2014; Zhou et al., 2016), which is straightforward in demonstrating the attention of each module in the model. However, visualization based methods are lack of theoretical support and highly dependent on human subjective understanding, which can be ambiguous. Other preliminary works focus on input attribution (Chattopadhay et al., 2018; Koh & Liang, 2017; Li et al., 2016; Selvaraju et al., 2017; Sundararajan et al., 2017), trying to distinguish the influential

---

[*]Correspondence to Kan Ren.

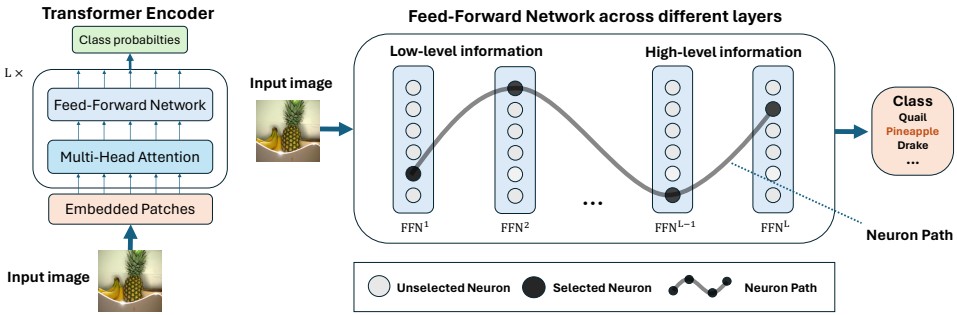

Figure 1: The illustration of the main concept of our work, focusing on the feed-forward network (FFN) component within a standard ViT (Dosovitskiy et al., 2021) encoder. In the left part, a typical ViT encoder is depicted, consisting of totally $L$ Transformer layers. The right part illustrates the neuron path discovered by our method, which identifies a path comprising of the neurons within the FFN module across the model layers. Each FFN in the encoder is denoted as $\text{FFN}^l$, $l \in [1, L]$.

region of the input. However, input explanation heavily relies on input images and cannot actually reveal the interior mechanism of the target model. Nowadays, discovering and explaining the important neuron in one trained model has drawn more and more attention. The related works either explain the patterns obtained at different network neurons via visualization (Foote et al., 2023; Ghiasi et al., 2022; Karpathy et al., 2015; Vig, 2019; Zeiler & Fergus, 2014), or study the effects of individual neurons (You et al., 2025; Dai et al., 2022; Dalvi et al., 2019; Durrani et al., 2020; Huang et al., 2023). Nevertheless, most of these methods overlook the correlation between the neurons in different layers and have not considered the complex joint influence of various neurons.

To recognize and understand the underlying mechanisms of multiple neurons across layers in state-of-the-art vision models, new explanation methods are needed. Inspired by the concept of visual pathway in neuroscience (Gupta et al., 2020), our work deliberately concentrates on uncovering the influential *neuron path* within the vision Transformer models, which is a sequence of neurons within each layer tracing from input to model output, to elucidate their significance and role in model inference. Details of neuron path are shown in Figure 1. To evaluate the importance of a set of neurons, we initially define a comprehensive joint influence measure for the target model. Using this measure as a guide, we demonstrate our approach using the image classification task (Deng et al., 2009; Dosovitskiy et al., 2021; He et al., 2022), illustrating how we identify the most influential neuron path that drive the model inference through a layer-progressive neuron locating algorithm.

We have conducted several quantitative and qualitative experiments on the found neuron path, illustrating the significant role it plays and the advantage of our solution discovering and explaining the critical part of vision Transformer models. The initial discovery is about the effectiveness of our proposed method and the critical impact that neuron path exerts to the model inference. What follows is that common influential neuron paths exist in the target model for images within the same category, and categories with similar semantic information also exhibit relatively high similarity in their neuron paths. Furthermore, the discovered neuron paths offer potential insights for model pruning and specialization in Vision Transformer models.

In summary, the contribution of our work is as follows. (i) We have proposed a novel method to reveal the crucial information transmission flow within vision Transformer model, based on locating and analyzing internal neurons as neuron path. (ii) We have performed a series of experiments, not only demonstrating the critical influence on the model inference brought by the discovered neuron paths, but also revealing the intrinsic potential mechanism of vision Transformer models on processing semantic information. (iii) We apply the identified neurons to guide model pruning, showing that vision Transformers exhibit redundancy and that the essential components are sparse.

## 2 RELATED WORK

### 2.1 VISION TRANSFORMER MODELS

In computer vision, various vision Transformer models (Caron et al., 2021; Dosovitskiy et al., 2021; Oquab et al., 2023; Radford et al., 2021) have showcased superior performance in real-world tasks

such as image classification (Dosovitskiy et al., 2021). Transformer-based encoders typically consist of multiple structural identical layers, each comprising a self-attention module and a feed-forward network (FFN). Typically, vision Transformer operates by dividing input images into fixed-size patches, treating them akin to tokens in natural language processing (NLP) (Vaswani et al., 2017), incorporating of positional encodings, a classification token, and processing them through self-attention mechanisms with subsequent FFNs. While vision Transformers have achieved excellent performance , the underlying mechanisms that contribute to their effectiveness remain opaque. Several recent works have attempted to unravel the inner workings of vision Transformer by attributing individual neurons to different concepts using language models (Guertler et al., 2023; Hernandez et al., 2021) or multimodal model (Oikarinen & Weng, 2023) , visualizations (Chefer et al., 2021; Ghiasi et al., 2022) or by observing inner behaviors or properties (Naseer et al., 2021; Paul & Chen, 2022; Zimmermann et al., 2024). However, the complexity of vision Transformer's architecture and the interactions between its blocks pose challenges in fully comprehending its capabilities. Therefore, our work seeks to explore the inner working of vision Transformer by investigating the role of neuron paths, capturing essential patterns of information flow within the model.

## 2.2 EXPLAINABILITY METHODS IN VISION MODELS

**Visualization based Methods.** Visualization-based methods are pivotal in interpreting deep neural networks used in vision tasks. These techniques primarily include methods like activation maximization (Erhan et al., 2009), saliency maps (Selvaraju et al., 2017), and class activation maps (CAMs) (Zhou et al., 2016). Activation maximization involves optimizing input images to maximize the activation of specific neurons, thereby providing insights into what features activate particular neurons. Saliency maps compute gradients of the output with respect to the input image to highlight regions contributing most to the network's prediction. CAMs uses the weighted sum of feature maps to visualize class-specific regions in the input image. Despite their usefulness, these methods face limitations. They can be computationally intensive and may produce unrealistic images that do not correspond to natural inputs (Nguyen et al., 2016). Further, they often suffer from noise and lack spatial localization, potentially leading to misleading interpretations (Smilkov et al., 2017). And these methods rely on specific network architectures and may not generalize well across different model types or tasks (Selvaraju et al., 2017). Visualization based methods represent a distinct perspective on explainability other than ours. They are more inclined to intuitively display the concerns of the visual model and succinctly represent the learning results of the model, whereas our focus is on the intrinsic mechanism of the model and an in-depth examination of the model's operational mechanism from input to output.

**Neuron based Methods.** The emergence of powerful deep learning models has stimulated interest in the study of model explainability. The concept of a neuron in machine learning, as discussed in various works (Bau et al., 2017; Dhamdhere et al., 2019; Oikarinen & Weng, 2023), encompasses hidden units, hidden layers, and individual latent variables. These elements are often considered fundamental computational components that process input information. Among the explainability approaches, neuron-based methods, a more intricate form of analysis (Fan et al., 2024), are typically divided into gradient-based and non-gradient based methods. Non-gradient based methods generally contain three approaches: neuron visualization (Foote et al., 2023; Karpathy et al., 2015; Li et al., 2016; Zeiler & Fergus, 2014), concept-based methods (Bau et al., 2017; Hernandez et al., 2021; Kim et al., 2018; Oikarinen & Weng, 2023; Panousis & Chatzis, 2024) and statistic-based methods (Ghorbani & Zou, 2020; Kwon & Zou, 2022; Shan et al., 2021; Yuan et al., 2021). However, these methods either rely on a data set of a close set of input-concept pairs (Bau et al., 2017), limiting their generalization ability to unseen concepts, or externally provided powerful models (Oikarinen & Weng, 2023; Panousis & Chatzis, 2024), such as CLIP (Radford et al., 2021), rather than intrinsic model properties. Further, some of their optimization goals focus on model output metrics, failing to fully elucidate the circulation of model knowledge and concepts. Gradient-based methods compute the attribution from output to input or target neurons (Chattopadhay et al., 2018; Dai et al., 2022; Dhamdhere et al., 2019; Selvaraju et al., 2017) via gradient computation or integrated gradient (Sundararajan et al., 2017). Though providing quantitative results for neuron explainability, they tend to treat all neurons equally and struggle with comparing neurons across different layers with the gradient flow numerically equitably, despite variations in depth. To address this issue, (Lu et al., 2020; 2021) try to discover a path of neurons rather than a cluster. However, all these methods fail to consider the joint influence of neurons across layers. It will not only result in biased outcomes due to

the domination of a few abnormally prominent neurons, but it will also fail to consider the coherence of information transmission, which makes it challenging to truly explore the intrinsic mechanism of the model. The objective of this paper is to propose a new method that addresses this issue.

In response to the limitations of existing methods, particularly those gradient-based methods focusing on individual neurons or input features, we introduce a novel approach for interpreting vision Transformer models (Dosovitskiy et al., 2021). Our method shifts the focus towards the collective contribution of selected neurons across different layers, offering a more comprehensive understanding of model behavior. Our method enables us to uncover influential pathways within the model, enhancing its explainability and providing valuable insights into its decision-making process.

# 3 NEURON PATH

As demonstrated above, vision Transformers (Dosovitskiy et al., 2021) exhibit complex structures and designs that present challenges for explainability. Research on vision Transformers predominantly focuses on visualization techniques related to attention maps (Caron et al., 2021; Oquab et al., 2023), with limited exploration of their neuronal anatomy. While in recent studies, efforts have been made to associate *individual* neurons with various concepts using language or multimodal models (Guertler et al., 2023; Hernandez et al., 2021; Oikarinen & Weng, 2023) and visualization techniques (Chefer et al., 2021; Ghiasi et al., 2022), the understanding of vision Transformer's internal neuronal organization remains relatively unexplored. Some related studies (Dai et al., 2022; Geva et al., 2021; Mitchell et al., 2021) have suggested that the FFN component of Transformer models can function as a memory mechanism due to its structural similarity to the attention mechanism and they encode human-interpretable, high-level concepts, which motivated this work to focus on these neurons in our experiments. Based on this insight, it is reasonable to hypothesize that knowledge or information related to visual patterns in vision Transformers may also be stored within the FFN. As stated before about the definitions of a neuron (Bau et al., 2017; Dhamdhere et al., 2019; Oikarinen & Weng, 2023), we focus on a specific type of neuron within the vision Transformer which is the activation of the output from the first linear layer in the FFN module of each Transformer block.

## 3.1 PRELIMINARY STUDY

We first conduct a preliminary study on neuron importance analysis for vision Transformer models to illustrate the motivation of our method. The existing works studying neuron activities of vision model inference (Dhamdhere et al., 2019; Sundararajan et al., 2017) mainly focus on convolutional neural networks (CNNs), trying to analyze convolution filter's sensitivity to the input. To assess the importance of neurons themselves within pretrained models, the concept of knowledge attribution (Dai et al., 2022) offers valuable insights on finding the individual neurons storing model knowledge such as "capital of country" in Transformer-based language models like BERT (Kenton & Toutanova, 2019). Inspired by this method, we analyze the effects of individual neurons in vision Transformer model contribution to the model prediction, with most influential neurons marked as "knowledge neurons". The details about the implementation of knowledge attribution can be found in Appendix B. Note that, it focuses solely on individual knowledge neurons, overlooking the joint effects of the common neurons.

By applying the method to two types of vision Transformer models using different pretrain paradigms, supervised ViT (ViT-B-16) (Dosovitskiy et al., 2021) and self-supervised Masked AutoEncoder (MAE-B-16) (He et al., 2022), we have derived an unex-

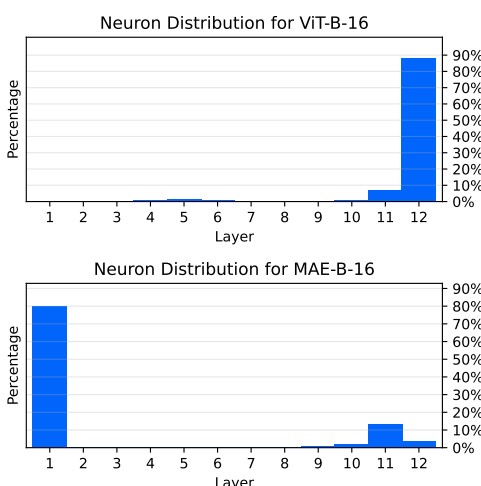

Figure 2: The distribution of knowledge neurons in two different pretrained vision Transformer models. It can be noticed that comparing ViT-B-16 (Dosovitskiy et al., 2021) and MAE-B-16 (He et al., 2022), their neuron attribution show completely opposite distributions across layers.

pected discovery, as illustrated in Figure 2. Despite both models being pretrained and finetuned on the same dataset (ImageNet (Deng et al., 2009)) with almost identical model structures, i.e., 12 Transformer layers with the same hidden size, and utilizing the same input process, i.e., 16-size patching on the input, we observe that *the distributions of individual knowledge neurons across the two models are almost inverted.* Moreover, numerous studies have demonstrated that, in vision models, information is progressively aggregated from low-level visual patterns to high-level visual concepts as the network depth increases (Bau et al., 2017; Zeiler & Fergus, 2014). This implies that, to accurately capture high-level concepts in an image, each layer should include at least one significant neuron to facilitate the flow of information. However, current attribution methods may fall short in fully capturing this requirement. Consequently, there is a pressing need for a new method that can provide deeper insights into the transmission of information by neurons within the model. It is true that the attribution of individual neurons using integrated gradients, as shown in Figure 2, yields clusters of neurons that are most sensitive to image information in the visual model, but this does not capture the process of how visual information is transmitted and processed through neural pathways. Just as that in the human visual system, nerve cells close to the retina and visual cortex must be the most sensitive, but the visual neural pathways between the two are equally important and worthy of study. Thus, a joint measurement of neuron path within the model is demanding.

## 3.2 Joint Attribution Score

To locate the most influential neurons, we first propose a joint influence measurement of a set of different neurons. Motivated by the concept of information flow (Lu et al., 2021), we introduce a novel integrated gradient-based method to compute the joint attribution of all selected neurons, across different layers in vision Transformer models. Consider a vision Transformer model $F$ consisting of $L$ layers. Given an input-label pair $< x, y >$, where $x \in \mathbb{R}^d$ and $y$ is the related label, we begin by denoting the output result utilizing the neurons within the first $N$ layers as

$$F_x(\hat{w}_{i_1}^1, \hat{w}_{i_2}^2, ..., \hat{w}_{i_N}^N) = p(y|x, w_{i_1}^1 = \hat{w}_{i_1}^1, w_{i_2}^2 = \hat{w}_{i_2}^2, ..., w_{i_N}^N = \hat{w}_{i_N}^N) , \qquad (1)$$

where $w_i^l$ represents the $i$-th intermediate neuron in the FFN module of the $l$-th layer ,with $1 \leq l \leq N \leq L$ and $\hat{w}_i^l$ is the assigned value of $w_i^l$. To calculate the contribution to the model output of the selected neurons jointly, we incrementally sum the values of $\{w_{i_1}^1, w_{i_2}^2, ..., w_{i_L}^L\}$ from 0 to their original values $\{\overline{w}_{i_1}^1, \overline{w}_{i_2}^2, ..., \overline{w}_{i_L}^L\}$ using a control factor $\alpha \in (0, 1)$. This process yields $L$ differentiations with respect to each neuron in the neuron path. By aggregating these differentiations and integrating with respect to $\alpha$, we obtain a score that represents the joint attribution of the neuron in the model. Formally, the definition of this joint attribution score is given as follow.

**Definition 1 (Joint Attribution Score)** *Given a model $F : \mathbb{R}^d \to \mathbb{R}$ containing $L$ layers, whose output with input $x$ is defined as $F_x$, with a set of neuron $\{w_{i_1}^1, w_{i_2}^2, ..., w_{i_N}^N\}, N \leq L$, a Joint Attribution Score is defined as*

$$JAS(w_{i_1}^1, w_{i_2}^2, ..., w_{i_N}^N) = \sum_{n=1}^{N} \overline{w}_{i_n}^n \int_{\alpha=0}^{1} \sum_{l=1}^{N} \frac{\partial F_x(\alpha \overline{w}_{i_1}^1, \alpha \overline{w}_{i_2}^2, ..., \alpha \overline{w}_{i_N}^N)}{\partial w_{i_l}^l} d\alpha . \qquad (2)$$

For the convenience of computation, we use the Riemann approximation to estimate the continuous integral as follows,

$$\widetilde{JAS}(w_{i_1}^1, w_{i_2}^2, ..., w_{i_N}^N) = \frac{1}{m} \sum_{j=1}^{N} \overline{w}_{i_j}^j \sum_{k=1}^{m} \sum_{l=1}^{N} \frac{\partial F_x(\frac{k}{m} \overline{w}_{i_1}^1, \frac{k}{m} \overline{w}_{i_2}^2, ..., \frac{k}{m} \overline{w}_{i_N}^N)}{\partial w_{i_l}^l} , \qquad (3)$$

where $m$ is the sampling step.

In Eq. (2), we extend and calculate integrated gradient, the cumulative gradient of the measure of interest (classification output) $F_x$ to the path of the target component (the neuron set $\{w_{i_1}^1, w_{i_2}^2, ..., w_{i_N}^N\}$) ranging from zero effect ($\alpha = 0$) to full effect ($\alpha = 1$), given the model input $x$. This assesses the full contribution of the target neuron set.

With Definition 1 stated above, we propose a novel neuron-based model analysis method *Neuron Path*, trying to find a path consisting at least one important neuron at each layer tracing from input to model output to elucidate their significance and role in model inference. The formal definition is as below.

**Definition 2 (Neuron Path)** *Given a model $F : \mathbb{R}^d \to \mathbb{R}$ containing $L$ layers, with an input $x$, and a user-defined criterion $\mathbf{S}(\cdot)$, a neuron path $\mathcal{P}_x$ is defined as follow.*

$$\mathcal{P}_x = \{w^1, w^2, ..., w^L\} \tag{4}$$

*that maximizes the $\mathbf{S}(\mathcal{P}_x)$, where $w^l, l \in \{1, 2, ..., L\}$ stands for the selected neuron within layer $l$.*

$\mathbf{S}(\cdot)$ utilizes a user-defined score to evaluate the behavior of the network with respect to the input of interest $x$. When the score function is defined as the maximum activation, the path $\mathcal{P}_x$ comprises the neurons with the highest activation in each layer, as detailed in Appendix C.2. When Pattern Influence (Lu et al., 2021) is applied, we obtain the path $\mathcal{P}_x$ that maximizes the corresponding influence score, as detailed in Appendix C.3. For our method, we use the JAS in Eq. (2).

### 3.3 Locating Influential Neuron Path

To identify the neuron path within the target model $F$ that maximizes the JAS in Definition 1, we employ a layer-progressive neuron locating algorithm, to iteratively select neurons layer by layer, gradually constructing the path that maximizes JAS. Utilizing the definition of the JAS provided above, we formulate the algorithm as follows. For a vision Transformer model comprising $L$ layers, we enumerate each layer and add neurons to the path to maximize JAS at the current depth. The detailed procedure of our algorithm is outlined in Algorithm 1. Time complexity analysis can be found in Appendix A.

---

**Algorithm 1** Layer-progressive Neuron Locating Algorithm

---

**Input**: Model $F$ with $L$ layers, input sample $x$
**Output**: neuron path $\mathcal{P}$
**Initialization**: $\mathcal{P} = \varnothing, l = 1$
**while** $l \leq L$ **do**
    $\mathcal{W}$ is the set of neurons in layer $l$ of $F$; Score $= 0, p =$ None
    **for** $w \in \mathcal{W}$ **do**
        **if** Score $< \widetilde{\text{JAS}}(\mathcal{P}, w)$ **then**
            Score $= \widetilde{\text{JAS}}(\mathcal{P}, w); p = w$
    $\mathcal{P} = \mathcal{P} \cup \{p\}; l = l + 1$

---

Comparing with existing methods that focus on individual neurons or input features (Dai et al., 2022; Dalvi et al., 2019; Durrani et al., 2020; Huang et al., 2023; Sundararajan et al., 2017), our approach considers the collective contribution of selected neurons across different layers in the model. By computing and maximizing the joint attribution scores, we aim to identify the most influential neuron path for the given input, which processes and conveys the most crucial information as flow through the model layers. Some related works (Lu et al., 2020; 2021) on Transformer-based language models also investigate neuron pathways, but they overlook the collective influence of neurons across layers, resulting in suboptimal outcomes, as illustrated in the experiment. In contrast, our approach offers several advantages. By considering the collective contribution of neurons across layers and employing a layer-progressive neuron locating algorithm, we can effectively identify neuron paths that maximize the crucial information flow. Furthermore, our method provides a holistic view of the model's information transmission during inference, offering valuable insights into its inner workings.

## 4 Experiments

In this section, we present both quantitative and qualitative experiments to analyze the proposed method from multiple perspectives. In Section 4.2, we validate the neuron paths discovered by our approach by comparing them with baseline methods through quantitative analysis and intervention experiments. In Section 4.3, we provide in-depth statistics on the discovered neuron paths and reveal their clustering patterns with respect to image classes. Finally, in Section 4.4, we perform network pruning on other neurons while preserving the identified neuron paths, offering insightful observations on classification performance and highlighting potential applications of our method. The code will be released upon the acceptance of this paper.

## 4.1 Implementation Details

Our experiments are conducted on ViT (Dosovitskiy et al., 2021) and MAE (He et al., 2022), two widely used vision Transformer models but with different pretraining methods. For our following experiments, we will mainly utilize 3 ViT settings and 1 MAE setting: **ViT-B-16**, **ViT-B-32**, **ViT-L-32** and **MAE-B-16** as the target models. Details of these models are in Appendix C.1.

These models are all trained for the image classification task, they are all pretrained on ImageNet21k and finetuned on ImageNet1k (Deng et al., 2009), whose validation set shall be the probing dataset for the experiments below. As for the calculation of JAS, we set the sampling step $m = 20$ in Eq. (3) for the following experiments.

## 4.2 Quantitative Comparison

In this section, we evaluate different neuron explainability methods and perform a neuron intervention experiments according to the results of different methods. We start by presenting the different methods of comparison, the first one is commonly utilized in literature (Dhamdhere et al., 2019; Dai et al., 2022) and the second one is presented in previous work (Lu et al., 2021). Details of the first two baseline methods implementation can be found in Appendix C.2 and C.3.

- **Activation** represents the method that locates the neuron with the largest activation at each layer.

- **Influence Pattern** follows the setup of the previous work (Lu et al., 2021), which utilizes the input to build path integral and find the neuron with largest integrated gradient layer by layer.

- **Neuron Path** (ours) stands for the neurons sampled by the neuron localization method we have proposed above in Section 3.3.

**Experimental settings.**   We assess the significance of neurons identified by different methods to evaluate the effectiveness of the compared approaches. The first experiment measures the joint influence of the neurons given different inputs to the model, as defined by the *Joint Attribution Score (JAS)* in Section 3.2.

Additionally, we will conduct a more in-depth comparison of the three methods through manipulation experiments by intervening in the identified neurons. These experiments involve two operations on the selected neurons: removal (zeroing out) and enhancement (doubling) on the values of the selected neurons. For each image input, neurons in the target model are identified using the three approaches, and manipulation experiments are conducted. By analyzing the changes in model outputs after these operations, we can observe in finer detail the impact of these neurons on the model's classification ability. Two metrics, *Removal Accuracy Deviation* and *Enhancement Accuracy Deviation*, are used to quantify the experimental results of the operations. These metrics measure the change in model accuracy after the corresponding operation. Further details about these metrics are provided in Appendix C.5. Moreover, we statistically analyze the deviations of the predicted probability regarding the ground-truth image class after performing the two operations. Additional details regarding the probability deviation metrics can be found in Appendix C.4. The quantitative results are presented in Table 1, and the statistical analyses are presented in Figure 3.

| Metrics | Methods | Target Models | | | |
|---|---|---|---|---|---|
| | | **ViT-B-16** | **ViT-B-32** | **ViT-L-32** | **MAE-B-16** |
| **Joint Attribution Score** ↑ | Activation | -0.0034 | 0.0288 | -0.0065 | 0.0013 |
| | Influence Pattern | 0.0412 | 0.0841 | 0.1227 | 0.0030 |
| | Neuron Path (ours) | **0.4078** | **0.6610** | **1.0086** | **0.0095** |
| **Removal Accuracy Deviation** ↓ | Activation | 0.07% | -0.15% | 0.16% | -2.80% |
| | Influence Pattern | -0.50% | -1.24% | -1.41% | -15.67% |
| | Neuron Path (ours) | **-2.40%** | **-3.81%** | **-5.28%** | **-26.50%** |
| **Enhancement Accuracy Deviation** ↑ | Activation | -0.33% | -0.45% | -0.86% | -1.00% |
| | Influence Pattern | 0.46% | 0.83% | 1.12% | 4.15% |
| | Neuron Path (ours) | **2.04%** | **3.06%** | **5.02%** | **7.28%** |

Table 1: Three different metrics were used to measure the neurons obtained using three different methods. ↑ (↓) means that the higher (lower), the better. The best results are in bold font.

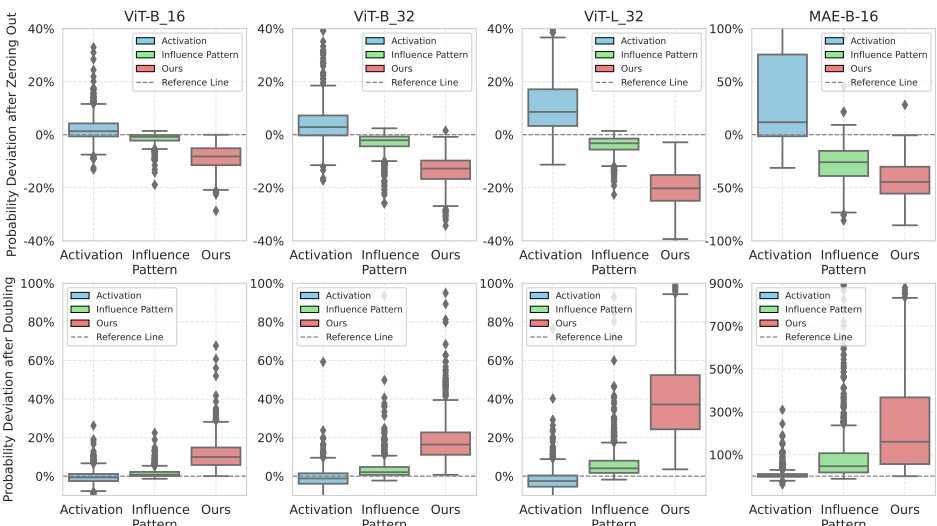

Figure 3: The relative deviation in the model's predicted probability of the ground-truth label when the value of neurons selected by different methods is either removed (zeroed out) or enhanced (doubled).

**Finding 1: The Neuron Path method more effectively identifies the influential neurons within the model.** In Table 1, our proposed Neuron Path method demonstrates a superior ability to identify neurons that have a more significant impact on model outputs, as indicated by larger JAS values. This is expected, as our method directly maximizes the joint contribution of the selected neurons. We further quantify the impacts of the neurons discovered by the Neuron Path method and other compared methods by measuring the deviation in model accuracy of classification when manipulating the value of neurons identified by different methods. As shown in the lower part of Table 1, the removal and enhancement of neurons identified by our Neuron Path method result in significantly larger drops and improvements in model performance, illustrating the effectiveness.

**Finding 2: The discovered neuron paths play a vital role in model inference .** Above results imply that the intrinsic neurons of the model significantly affects prediction performance. In order to explore in depth how the model's output probability is affected by the internal neurons, we count the relative deviation in prediction probability caused by manipulating the values of the discovered neurons. As illustrated in Figure 3, our method results in significantly larger positive contributions when enhancing neuron values, and more substantial negative effects when removing neurons comparing to other two methods. It indicates that neurons selected by our methods can significantly affect the model prediction, which demonstrates the crucial role these neurons play in model inference.

## 4.3 CLASS-LEVEL ANALYSIS

In this subsection, we aim to understand the internal workings of vision model predictions and determine if neuron paths reflect the intrinsic mechanisms of the model architecture that critically contribute to predictions. To this end, we analyze the common characteristics of neuron paths at the class level. Interestingly, this analysis reveals clustering properties within the same class and semantic relationships across different classes.

**Intra-class analysis on neuron paths.** We aggregated the neuron paths for images of the same category and compiled statistics on the frequency of each neuron selected by our proposed method. Figure 4 shows the frequency distribution of each neuron (vertical axis) across different layers (horizontal axis) in the discovered neuron paths for four different classes. The width of the violin plot indicates the frequency with which a neuron is selected. The most frequently used neuron at each layer is connected with a dotted line. Due to the restriction that a category only contains limited number of images, the length of the violin plot is shortened if the the ends of neuron index are not sampled. See Appendix D.1 for more details and results.

**Finding 3: Some certain neurons contribute more at each layer to specific classes.** From Figure 4, it is evident that at most layers, certain neurons are selected with significantly higher frequency

Class: albatross     Class: Weimaraner     Class: Border collie     Class: Eskimo dog

Figure 4: The frequency of each neuron at each layer occurred in the discovered neuron paths.

in neuron paths corresponding to images of the same class. These highly selective neurons exhibit clear clustering properties. This phenomenon reveals the intrinsic functioning of each layer in the vision model (ViT-B-16 (Dosovitskiy et al., 2021) in this experiment) in responding to inputs from specific classes.

**Inter-class analysis on neuron paths.** We extend our analysis to the inter-class level by examining the above stated neuron frequency distribution, which reflects neuron utilization in the neuron paths, to gain insights into the intrinsic working mechanisms of the vision model. We gather neuron utilization data — specifically, the frequency statistics of each neuron in the discovered paths — and construct a neuron utilization matrix $M_c \in \mathbb{R}^{L \times n}$, where $L$ is the number of layers and $n$ is the number of neurons per layer, for each class $c \in \mathcal{C}$. We then calculate the cosine similarity between $M_{c_1}$ and $M_{c_2}$ for two different classes $c_1, c_2 \in \mathcal{C}$. More details are in Appendix D.2.

**Finding 4: Neuron paths reveals semantic similarity.** The matrix $M_c$ reflects the neuron utilization of the neuron paths for image class $c$, highlighting the model's internal working properties for that class. As illustrated in Figure 5, semantically similar image classes exhibit similar neuron utilization patterns. This indicates that the discovered neuron paths tend to carry the semantic information corresponding to the category, suggesting that the vision model utilizes specific functional components in response to inputs from particular classes. It further implies that other than directly using visualization tools (Erhan et al., 2009; Selvaraju et al., 2017; Zhou et al., 2016) to subjectively illustrate the focus of model's each part, our method tends to localize the concept neurons straightaway. More examples can be found in Appendix D.3.

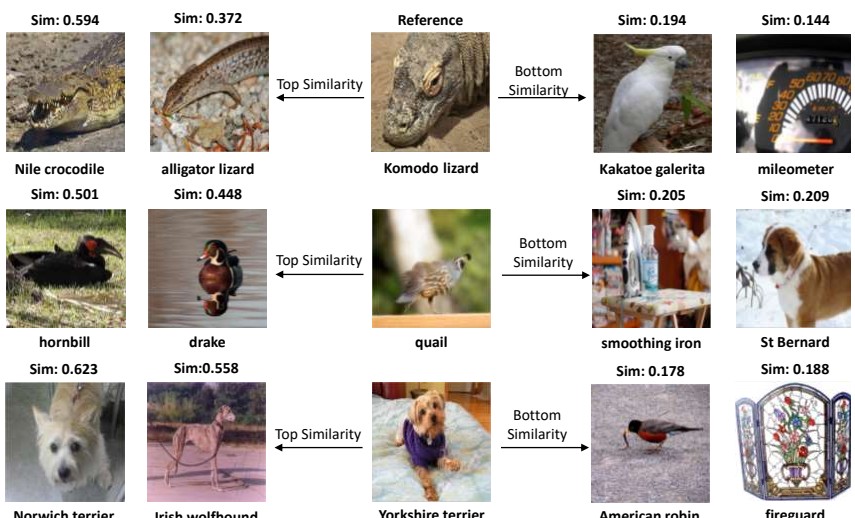

Figure 5: Examples of category similarity analysis. Using ViT-B-16 as target model, we randomly select three categories and calculate the similarity with others using the neuron utilization matrices and sample the top 5% and bottom 5% similar items. Through visualization we can see that categories with high (low) neuron path similarity tend to be also high (low) in semantic similarity.

## 4.4 MULTI-NEURON MODEL PRUNING

In this section, we explore potential applications of our neuron path method in model compression. Drawing inspiration from the Mixture-of-Depths approach (Raposo et al., 2024), we hypothesize that neurons in Vision Transformer models may exhibit redundancy. We propose that by retaining only

the most significant neuron paths, the model can sustain comparable performance, even when other neurons are randomly masked. To test this, we designed the following experiment. For a selected model, we retain the top $t \in \{1, 5, 10, 30, 50\}$ most influential neurons per layer within the neuron paths discovered by our Neuron Path method. Following the statistical procedure outlined in Section 4.3, we identified neuron paths for each category using the 80% of the image data, and transfer the results and conduct the pruning experiment on the rest 20% image data, establishing a generalization setting. During the pruning phase, we randomly zero out $p \in \{10\%, 30\%, 50\%, 100\%\}$ of the neurons, excluding the selected ones, to measure the effect on classification accuracy. The experimental results are presented in Figure 6, with experiment details and comparisons in Appendix E.

**Finding 5: Neurons within Vision Transformer models are largely redundant, with only a sparse subset significantly impacting model performance.** Our results in Figure 6 indicate that by retaining the critical neurons identified through our method, the model's performance remains robust, even when a significant percentage of other neurons are zeroed out. This implies that aside from a few key neurons, most neurons are either redundant or may even negatively affect the model's performance. Moreover, the results show that the number of important neurons is limited; the top five neurons deliver the best performance during pruning, while increasing the number of retained neurons leads to a decline in performance. This experiment also demonstrates the strong generalization ability of our approach. Also combining the results of important neuron value intervention in Section 4.2, we can conclude that our method can locate the most influential neurons conveying critical information flows across layers within the model.

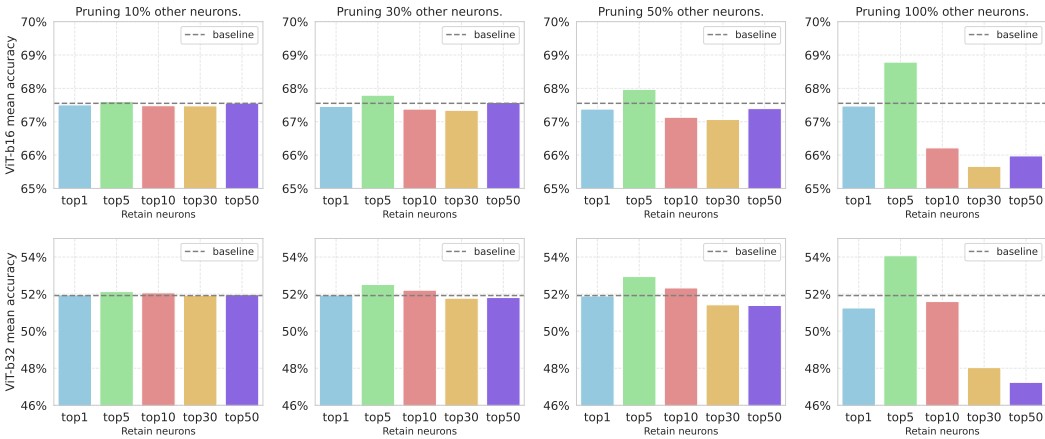

Figure 6: Model performance after different proportions of neuron pruning using ViT-B-16 and ViT-B-32. The dotted line represent the original performance of the used model, the y-axis represent the mean accuracy of the edited model and the x-axis represent the number of neuron in each layer we preserved. Baseline means the performance of the original model.

## 5 CONCLUSION AND FUTURE WORKS

In this study, we aim to unveil the intrinsic working mechanisms of vision Transformer models by locating and analyzing the crucial neurons at each layer that have the most significant impact on model inference. We introduce a novel approach called Neuron Path, guided by a newly proposed joint neuron attribution measure, to progressively identify the neurons that play key roles in information processing and transmission through model layers. Our series of analytical experiments reveal not only the importance of the discovered neuron paths in model inference but also illustrate some valuable insights into the internal workings of vision Transformers. Our work contributes to a more nuanced understanding of how these models manage visual information, expecting to advance the field of model explainability and facilitate future research on safely deploying vision models.

However, our method has certain limitations, which point to directions for future work. First, expanding the analysis beyond neurons in the FFN components to encompass the entire Transformer block could provide deeper insights into Vision Transformers. Second, exploring more downstream tasks like extending our approach from discriminative tasks to the segmentation and generative paradigm of vision models offers a promising avenue for future research.

## 6 ACKNOWLEDGEMENT

This research was supported in part by National Natural Science Foundation of China (Grant No. 62406193). The authors are also grateful for the support from Shanghai Frontiers Science Center of Human-centered Artificial Intelligence, MoE Key Lab of Intelligent Perception and Human-Machine Collaboration, and HPC Platform of ShanghaiTech University.

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

# A   LAYER-PROGRESSIVE NEURON LOCATING ALGORITHM ANALYSIS

Here is the estimation of the computing complexity of our proposed searching algorithm. We firstly define some notations of the model:

- $L$: Number of model layers
- $n$: Number of neuron in each layer
- $m$: Sampling step
- $d$: Feature dimension
- $T$: Transformer token number

In each layer, we will firstly iterate each neuron, whose complexity is $O(n)$, then for the calculation of JAS as Definition 1, the complexity of forward pass is $O(T \cdot d^2)$. And for backward propagation, the complexity is also $O(T \cdot d^2)$. Since we have $m$ sampling steps and $n$ neurons, the whole complexity is $O(m \cdot n \cdot T \cdot d^2)$, and for the addition operation, the complexity is $O(m \cdot n)$. Therefore, for one layer, the computing complexity is

$$O(N) + O(m \cdot n \cdot T \cdot d^2) + O(m \cdot n) \approx O(m \cdot n \cdot T \cdot d^2).$$

And for $L$ layers, the whole complexity is

$$O(L \cdot m \cdot n \cdot T \cdot d^2).$$

In the implementation, we can parallelly compute the integrated gradient, the detail of hardware and run time is shown in Appendix C.

## B    PRELIMINARY EXPERIMENT METHOD

Given a pretrained model $F$ with sample input $\mathbf{x}$, we can calculate the attribution score for the $i$-th neuron in the $l$-th layer $w_i^l$, which is called Knowledge Attribution (Dai et al., 2022), by the following formula

$$\text{Attr}(w_i^l) = \overline{w}_i^l \int_{\alpha=0}^1 \frac{\partial F_x(\alpha \overline{w}_i^l)}{\partial w_i^l} d\alpha \tag{5}$$

where $\overline{w}_i^l$ is the original value of the $i$-th neuron in layer $l$. For the sake of easy calculation, we use the Riemann approximation instead

$$\widetilde{\text{Attr}}(w_i^l) = \frac{\overline{w}_i^l}{m} \sum_{k=1}^m \frac{\partial F_x(\frac{k}{m}\overline{w}_i^l)}{\partial w_i^l} \ , \tag{6}$$

where $m$ stands for the sampling step, and for this experiment, we set $m = 20$. For each sample input, we can compute the attribution of each neuron and select the 5 neurons with the largest attribution as the significant neurons for that sample.

## C    QUANTITATIVE EXPERIMENT DETAILS

All the experiments are run on NVIDIA A40 GPUs with batch size equals to 10 and sampling step $m$ equals to 20, using ImageNet1k validation set. The estimated time for an experiment is about 10-20 hours based on the size of target model.

### C.1    TARGET MODEL SETTING DETAILS

ViT-B-16 stand for containing 12 Transformer blocks with patch size equals to 16; ViT-B-32 stand for containing 12 Transformer blocks with patch size equals to 32; ViT-L-32 stand for containing 24 Transformer blocks with patch size equals to 32 and MAE-B-16 stand for containing 12 Transformer blocks with patch size equals to 16. For ViT-B-16, ViT-B-32, MAE-B-16, their hidden sizes are all 768 and FFN inner sizes are all 3072, which stands for 3072 different neurons per layer; and for ViT-L-32, its hidden size is 1024 and FFN inner size is 4096, which stands for 4096 different neurons per layer.

### C.2    ACTIVATION

The activation method represents a path sampling technique that locates the neuron with the largest original activation value in each layer. The key idea is to trace the most activated neurons across layers, assuming that the most significant information flow is through these highly activated neurons. To find the Neuron Path $\mathcal{P}_x = \{w^1, w^2, ..., w^L\}$ in Definition 2, for an input sample $x$, we first initialize the path $\mathcal{P}_x$ as an empty set and then start a layer-wise neuron selection. For each layer $l$ from FFN of vision Transformer model, we compute the original values for all neurons in the layer, which is denoted as

$$A^l = \{\overline{w}_i^l\}, i \in \{1, 2, \ldots, n\} \ , \tag{7}$$

where $\overline{w}_i^l$ is the original value of the $i$-th neuron in layer $l$ and $n$ is the number of neurons per layer. To identify the neuron in each layer, we pick the neuron with the largest activation, and add this to the path and finally we obtain

$$\mathcal{P}_x = \{w^l | w^l = \arg\max_w(A^l)\}, l \in \{1, 2, \ldots, L\} \ . \tag{8}$$

After iterating through all layers, the path contains the most activated neurons from each layer, representing the route with the highest neuron activation. The resulting path highlights the neurons with the highest activation at each layer, suggesting a direct route of significant information flow through the network.

## C.3 INFLUENCE PATTERN

The Influence Pattern method, based on (Lu et al., 2021), utilizes the input to build path integrals and identifies the neuron with the largest integrated gradient in each layer. This method focuses on the influence of input features on neuron activations, tracing the path that maximizes the integrated gradient layer by layer. Their core algorithm is greedy search with key metric is Pattern Influence. With a target model $F$ with $L$ layers, we present

$$\mathcal{I}(x, \mathcal{P}_x) = \int_0^1 \prod_{l=1}^L \frac{\partial w^l(x' + \alpha(x - x'))}{\partial w^{l-1}(x' + \alpha(x - x'))} \, d\alpha \tag{9}$$

where $x'$ is the all zero base input, $w^l(\cdot)$ is the activation value of selected neuron of the $l$-th layer with respect to the input, $\mathcal{P}_x$ is the path containing the neurons across all the layers. With the definition above we can now form the greedy search-based algorithm as follows.

---

**Algorithm 2** Greedy Search-Based Influence Pattern Algorithm

---

**Input**: Model $F$ with $L$ layers, input sample $x$
**Output**: neuron path $\mathcal{P}$
**Initialization**: $\mathcal{P} = \varnothing, l = 1$
**while** $l \leq L$ **do**
   $\mathcal{W}$ is the set of neurons in layer $l$ of $F$; Score $= 0, p =$ None
   **for** $w \in \mathcal{W}$ **do**
     **if** Score $< \tilde{\mathcal{I}}(x, \mathcal{P} \cup \{w\})$ **then**
       Score $= \tilde{\mathcal{I}}(x, \mathcal{P} \cup \{w\}); p = w$
   $\mathcal{P} = \mathcal{P} \cup \{p\}; l = l + 1$

---

After iterating all the layers in model $F$, we will have a neuron path sampled based on Influence Pattern.

## C.4 PROBABILITY DEVIATION

Suppose we have a model $F : \mathbb{R}^d \rightarrow \mathbb{R}$, a neuron path $\mathcal{P}_x = \{w^1, w^2, ..., w^L\}$ where $w^l, l \in \{1, 2, ..., L\}$ stands for the selected neuron in layer $l$. Given a input pair $< x_i, y_i >$, we can have the output probability as

$$P_{x_i} = F_{x_i}(\hat{w}^1, \hat{w}^2, ..., \hat{w}^L) = p(y_i | x_i, w^1 = \hat{w}^1, w^2 = \hat{w}^2, ..., w^N = \hat{w}^L) , \tag{10}$$

We conduct manipulation experiments by manipulating the intermediate neurons within the neuron path, resulting in manipulated intermediate neurons $\tilde{w}^l$. The output probability with the manipulated intermediate neurons and the input sample $x_i$ are then

$$\tilde{P}_{x_i} = F_{x_i}(\tilde{w}^1, \tilde{w}^2, \ldots, \tilde{w}^L) \tag{11}$$

Then, the Probability Deviation with the input $< x_i, y_i >$ pair $\Delta P_{x_i} / P_{x_i}$ is defined as a ratio

$$\frac{\Delta P_{x_i}}{P_{x_i}} = \frac{\tilde{P}_{x_i} - P_{x_i}}{P_{x_i}} \tag{12}$$

The average Probability Deviation of the dataset is defined as the average of the individual probability deviations

$$\Delta P_{\text{average}} = \frac{1}{N} \sum_{i=1}^N \frac{\Delta P_{x_i}}{P_{x_i}} \tag{13}$$

The median Probability Deviation of the dataset is defined as the median of the individual probability deviations

$$\Delta P_{\text{median}} = \text{median} \left\{ \frac{\Delta P_{x_i}}{P_{x_i}} \mid \langle x_i, y_i \rangle \right\} \tag{14}$$

## C.5 Accuracy Deviation

Suppose we have a model $F : \mathbb{R}^d \to \mathbb{R}$, a neuron path $\mathcal{P}_x = \{w^1, w^2, ..., w^L\}$ where $w^l, l \in \{1, 2, ..., L\}$ stands for the neuron in layer $l$. Given a dataset $\mathcal{D}$ represented by $< x_i, y_i >$ pairs, where $x_i$ denotes the input of the $i$-th sample and $y_i$ represents its target label. The accuracy, denoted by Acc, is defined as

$$\text{Acc} = \frac{1}{|\mathcal{D}|} \sum_{i=1}^{|\mathcal{D}|} \mathbb{I}(\hat{y}_i = y_i) \tag{15}$$

where $\hat{y}_i$ is defined as

$$\hat{y}_i = \text{argmax}(P_{x_i}) \tag{16}$$

In this context, $P_{x_i}$ represents the output probability of the model. After manipulating the intermediate neurons within the neuron path as demonstrated in Appendix C.4, the output probability with the input sample $x_i$ become $\tilde{P}_{x_i}$, and $\tilde{y}_i$ changes to

$$\tilde{y}_i = \text{argmax}(\tilde{P}_{x_i}) \tag{17}$$

The accuracy also changes to

$$\widetilde{\text{Acc}} = \frac{1}{|\mathcal{D}|} \sum_{i=1}^{|\mathcal{D}|} \mathbb{I}(\tilde{y}_i = y_i) \tag{18}$$

Then, the Accuracy Deviation $\Delta \text{Acc}$ is defined as

$$\Delta \text{Acc} = \widetilde{\text{Acc}} - \text{Acc} \tag{19}$$

# D  MORE EXAMPLES AND IMPLEMENTATION DETAILS ABOUT CLASS-LEVEL ANALYSIS

## D.1  MORE FREQUENCY PLOT VISUALIZATIONS

In this section, we show more examples about the statistic visualization for four different models. In figures some violin plot may be shorter than others, this is due to the fact that, constrained by the limited number of image samples within a category, the length of the violin plot is shortened if the neurons at the ends of index are never selected in the neuron paths.

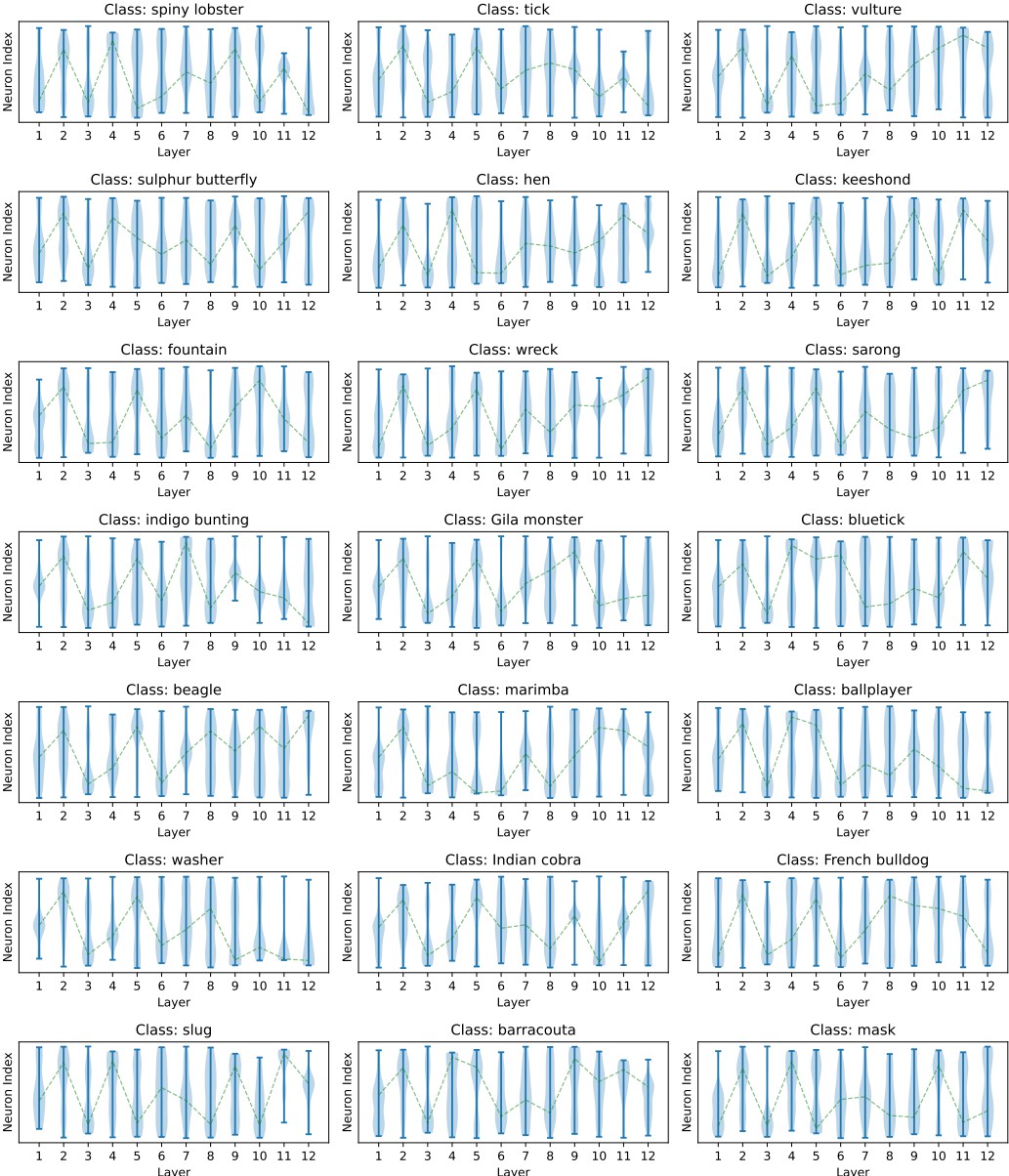

Figure 7: More Path Statistic Visualization for ViT-B-16

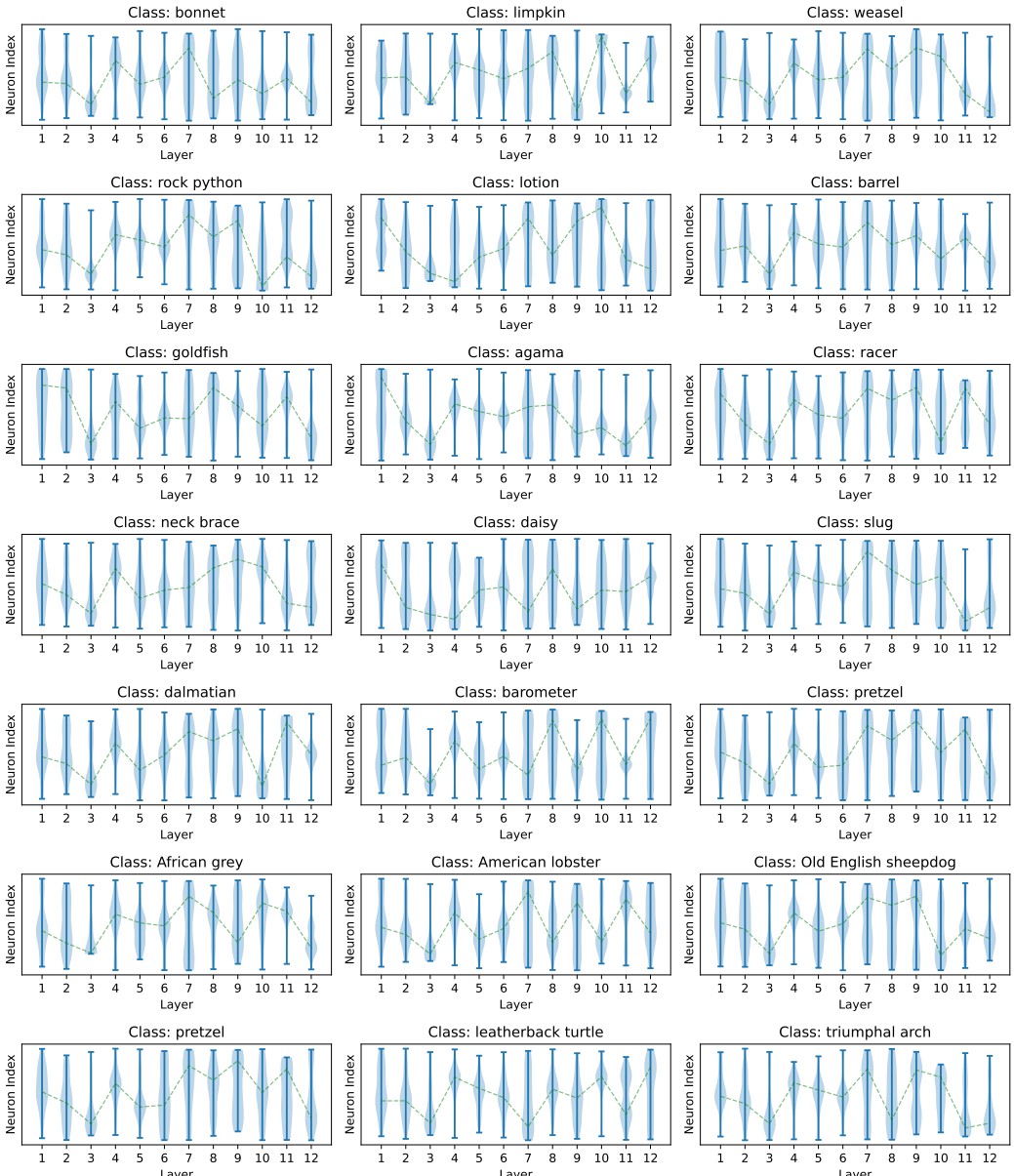

Figure 8: More Path Statistic Visualization for ViT-B-32

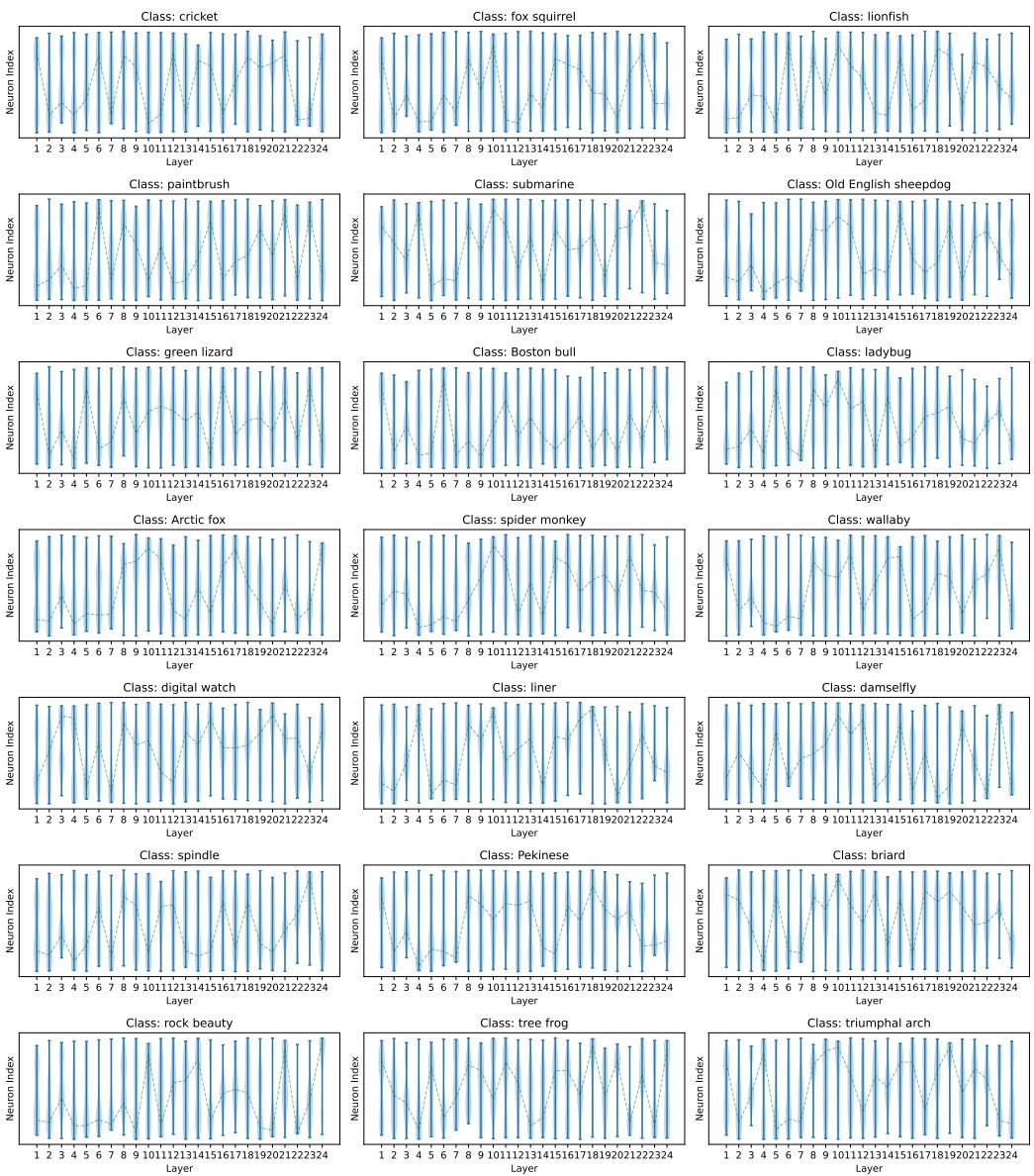

Figure 9: More Path Statistic Visualization for ViT-L-32

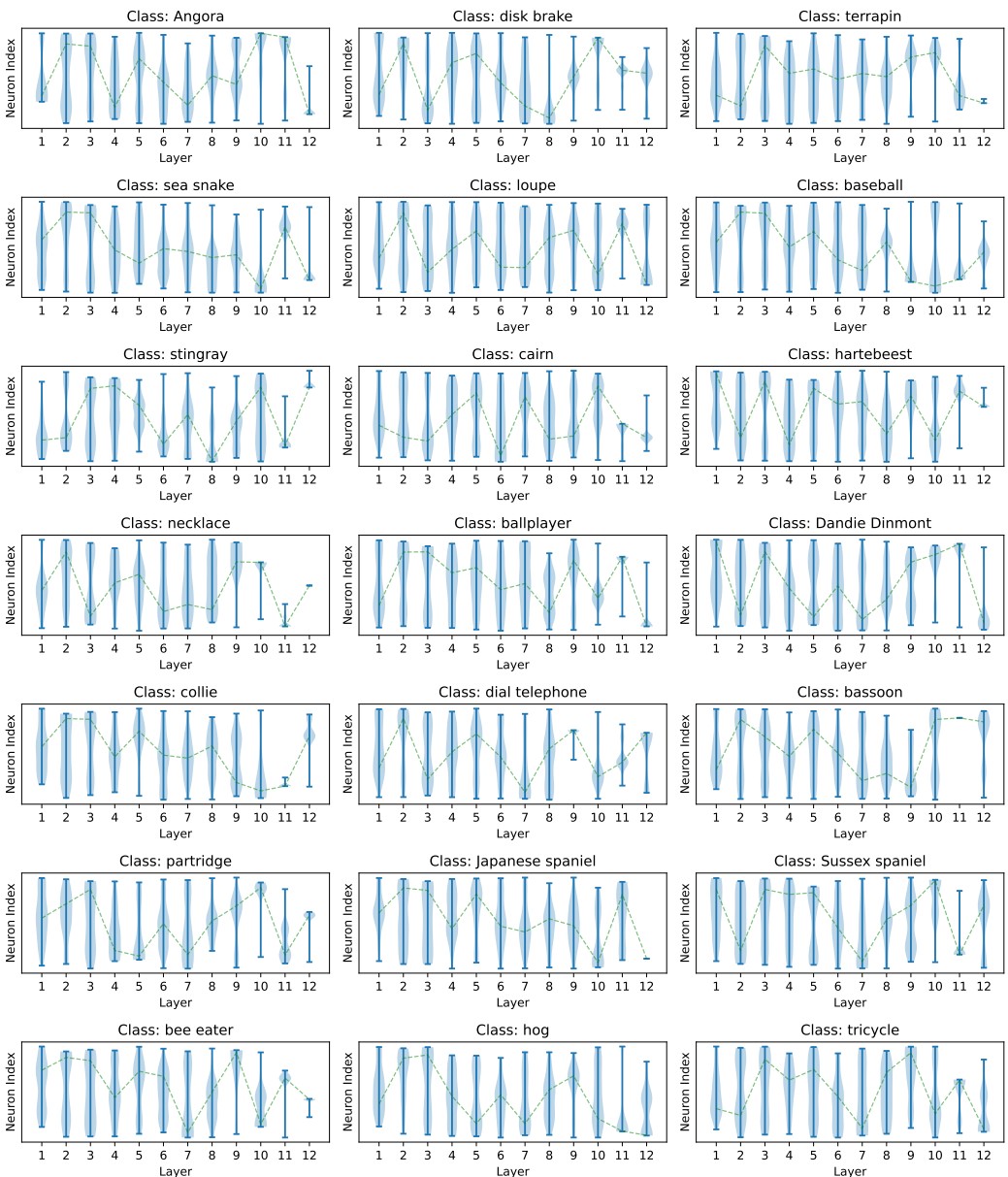

Figure 10: More Path Statistic Visualization for MAE-B-16

## D.2 NEURON UTILIZATION MATRIX

For a category $c \in \mathcal{C}$, it will contain $I$ different images, and in our setting, we use the ImageNet1k(Deng et al., 2009) validation set, so $I = 50$, $|\mathcal{C}| = 1000$. Each image will have its Neuron Path $\mathcal{P}_i$, so that we can have a matrix $f \in \mathbb{R}^{L \times n}$ to count the number of times each neuron index occurs in each layer, where $L$ represents the number of layers and $n$ stands for the number of neuron in each layer. By dividing the total number of neuron selections in each layer, we can define a neuron utilization matrix corresponding to category $c$ as

$$M_c \in \mathbb{R}^{L \times n} \; , \tag{20}$$

and for every pairs of neuron utilization matrix $M_{c_i}, f_{c_j}$, we can calculate their cosine similarity $s$ through following approaches.

$$s_{i,j} = \frac{M_{c_i} \cdot M_{c_j}}{||M_{c_i}|| \times ||M_{c_j}||} \; , \tag{21}$$

since we will have $|\mathcal{C}| \times |\mathcal{C}|$ pairs, we can have a similarity matrix $S \in \mathbb{R}^{|\mathcal{C}| \times |\mathcal{C}|}$, where $S_{i,j}$ represents the cosine similarity between $c_i$ and $c_j$, that is $s_{i,j}$

## D.3 MORE SEMANTIC SIMILARITY VISUALIZATION

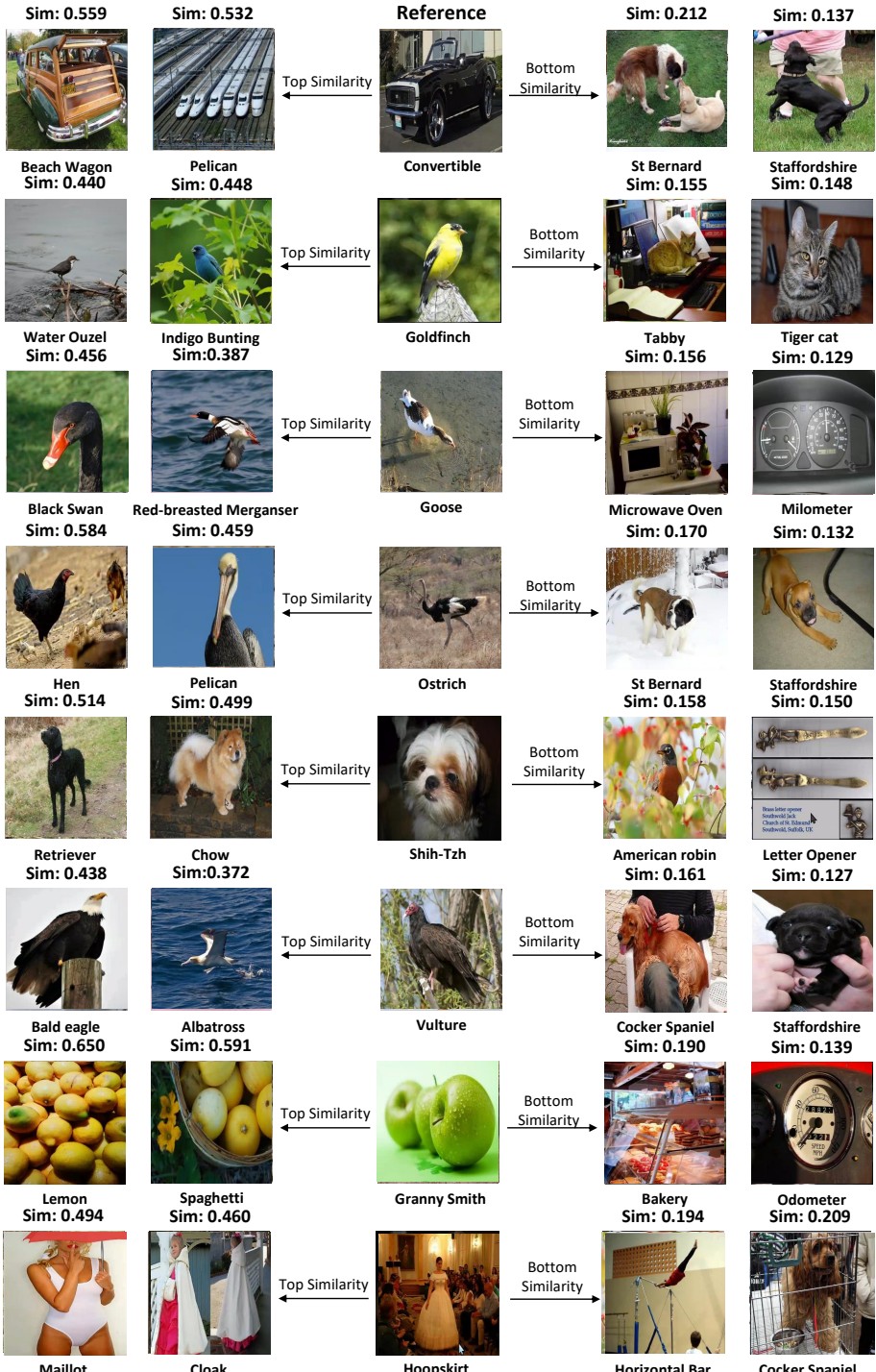

Figure 11: More examples of category similarity for ViT-B-16

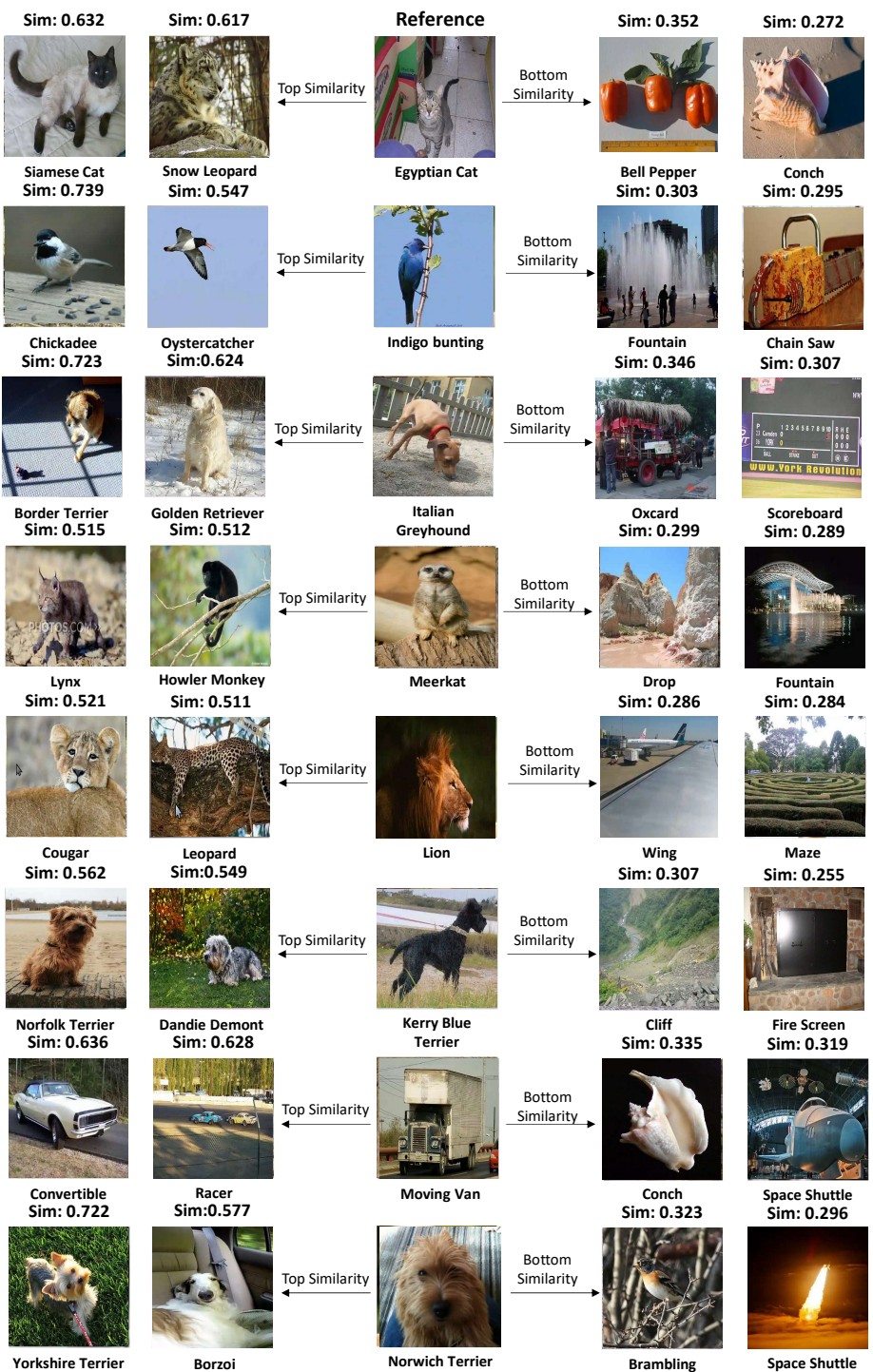

Figure 12: More examples of category similarity for ViT-B-32

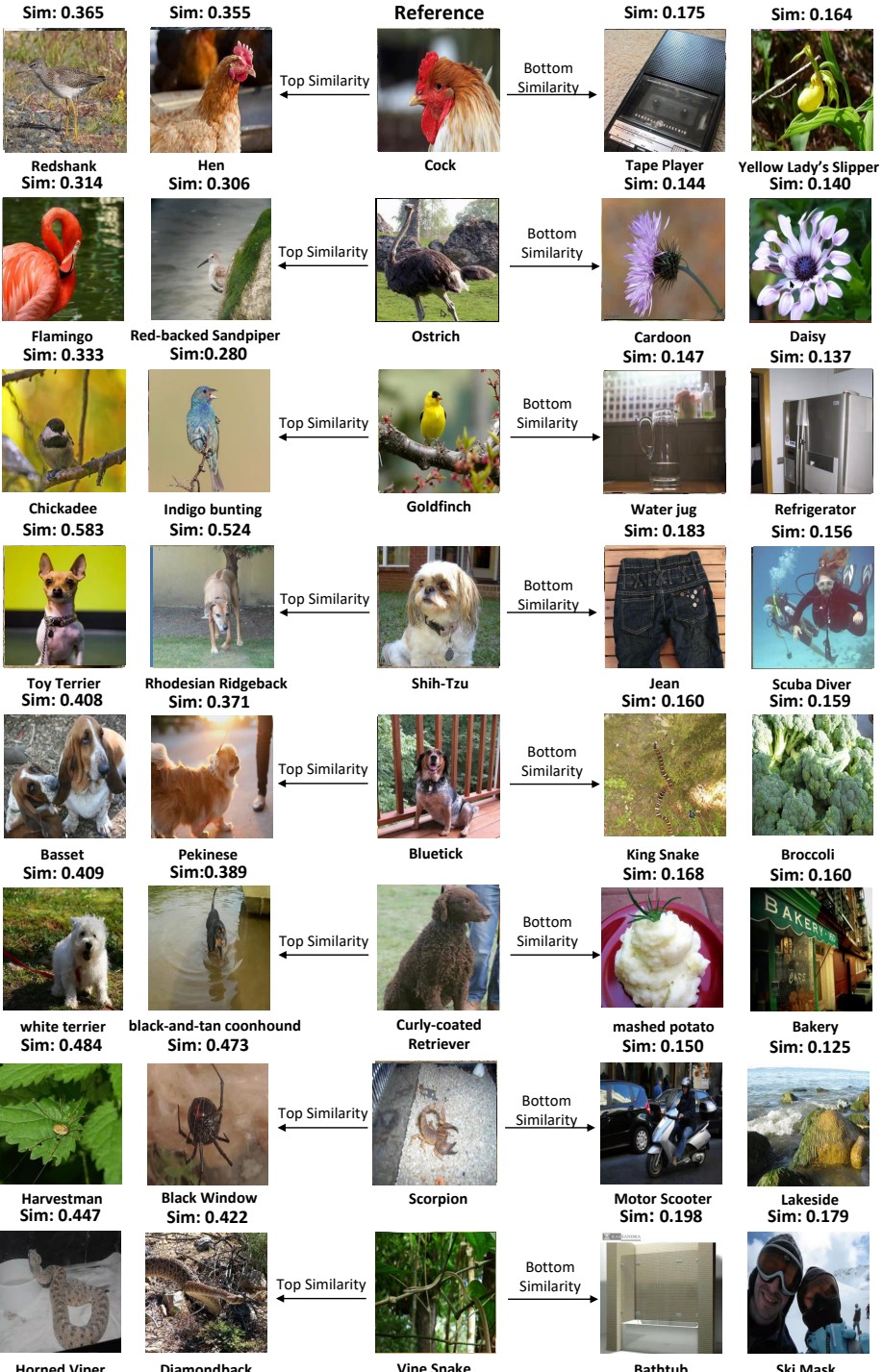

Figure 13: More examples of category similarity for ViT-L-32

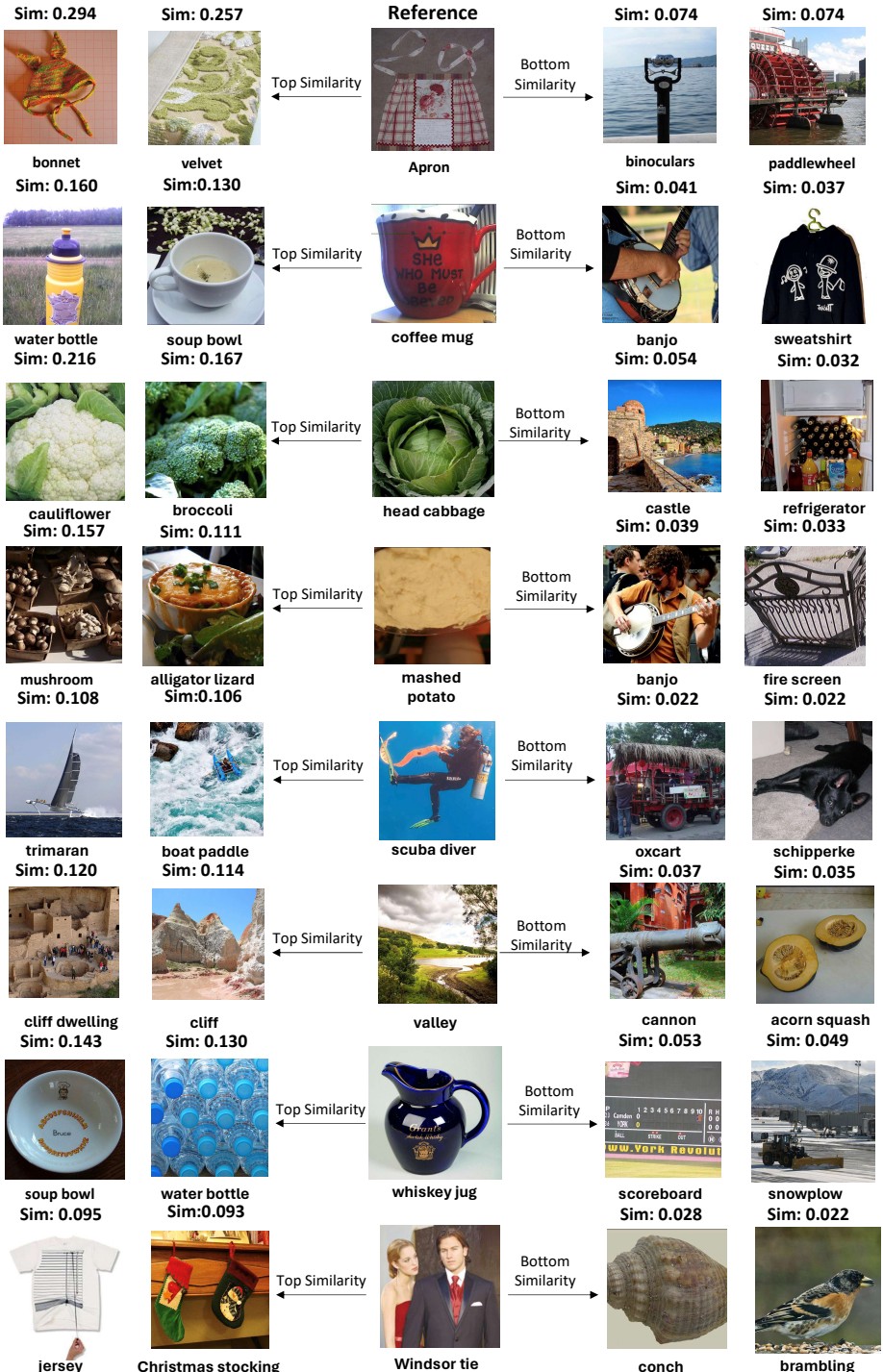

Figure 14: More examples of category similarity for MAE-B-16

# E    IMPLEMENTATION DETAILS OF MODEL PRUNING

## E.1    TopK neurons preserving algorithm

The calculation of accuracy change is using the same implementation of Appendix C.5. Instead of keeping only the neurons with highest JAS along the neuron path, we will retain the neurons with the TopK highest scores. The detailed procedure of our TopK neurons preserving algorithm is outlined in Algorithm 3.

---

**Algorithm 3** Layer-progressive Neuron Locating Algorithm Maintaining TopK Neurons

---

**Input**: Model $F$ with $L$ layers, input sample $x$, TopK$= t$
**Output**: neuron path $\mathcal{P}_t$
**Initialization**: $\mathcal{P}_t = \varnothing, l = 1$
**while** $l \leq L$ **do**
    $\mathcal{W}$ is the set of neurons in layer $l$ of $F$; Score $= \varnothing, p =$ None
    **for** $w \in \mathcal{W}$ **do**
        Score $= \text{TopK}_{K=t}(\widetilde{\text{JAS}}(\mathcal{P}_t, w))$
        $p = \arg(Score)$
    $\mathcal{P}_t = \mathcal{P}_t \cup \{p\}; l = l + 1$

---

## E.2    Probing and Test Set Splitting in Pruning Experiment

To evaluate the performance of the pruned models, we employ the ImageNet1k validation set (Deng et al., 2009), which comprises 50 images per class. For each class, 80% (40 images per class) are randomly selected to identify the TopK neurons in each layer. Specifically, for each of these 40 images, we apply the TopK neuron preserving algorithm described in Algorithm 3 to compute the top neurons. The results are then aggregated across all 40 images, yielding a set of $40 \times t$ neurons per class for each layer. By calculating the frequency of each neuron in this aggregated set, as detailed in Section 4.3, we determine the most frequently activated neuron paths. The neurons that appear most often are deemed the most critical for the respective class, and from this frequency distribution, the TopK neurons with the highest occurrence are selected as the representative top neurons for each layer. Subsequently, the pruning operation is executed using the remaining 20% of the dataset (10 images per class) as inputs. For these images, neurons not included in the selected TopK set are zeroed out in each layer.

This validation approach ensures that the pruning operation is tailored to each class by retaining the most relevant neurons based on their frequency in the validation subset. Evaluating the pruned model on the remaining images allows us to assess the strategy's effectiveness in preserving classification accuracy while significantly reducing model complexity.

## E.3    Quantitative comparisons of pruning methods

We tried using ViT-Slim (Chavan et al., 2022) as our compression baseline, here are the results.

| Method | Accuracy ↑ |
|---|---|
| **ViT-Slim** | $9.96\%$ |
| **JAS-Prune** | $67.7\%$ |

Table 2: The model prediction accuracy after conducting pruning. ↑ means that the higher, the better.

For fairness, we restrict ViT-Slim to focus solely on FFNs, mirroring our approach. Notably, while ViT-Slim emphasizes module weights, our method targets individual neurons. For dataset configuration, we employ the validation set split consistent with that described above in Appendix E.2. Although ViT-Slim may be more efficient, its performance is constrained by training on a limited dataset, a limitation partly attributable to its design.

Furthermore, we conduct ablation experiments on neuron path lengths to investigate their impact on the pruning effect. We also perform additional comparisons by employing activation values as neurons, using the same configuration as described in Section 4.2, wherein the TopK activation values in each FFN block are selected as neurons. In parallel. Here, **Depth** corresponds to the number of probing layers (i.e., the neuron path length), while **Mask** represents the percentage of neurons deleted, excluding the selected neurons.

| Depth | Mask (%) | Accuracy ↑ | |
| --- | --- | --- | --- |
| | | **Activation topK=5** | **JAS topK=5** |
| 6 | 100 | 0.517991 | 0.521252 |
| 9 | 100 | 0.518137 | 0.527151 |
| 12 | 100 | 0.531976 | 0.540702 |

Table 3: The model prediction accuracy after conducting pruning using different neurons and depth. ↑ means that the higher, the better.

The results demonstrate that our method outperforms the activation baseline, and that the depth of the neuron path is a also significant.

