# OpenReview forum: "Discovering Influential Neuron Path in Vision Transformers"
_ICLR.cc/2025/Conference — ICLR 2025 Poster_

### Official Review · Reviewer_K2QS · 2024-10-29

**Soundness:** 4
**Presentation:** 4
**Contribution:** 3
**Rating:** 6
**Confidence:** 4

**Summary:**

To discover the most influential neuron path, this paper propose a joint influence measure to assess the contribution of a set of neurons to the model outcome and a layer-progressive neuron locating approach. In the experiments section, this paper first conducted a quantitative comparison with two baseline methods: Activation & Influence Pattern; And then perform a  class-level analysis and pruning validation.

**Strengths:**

1. This paper is well-written and easily understood, especially for the `Introduction` and `Related Work` sections.

2. What's more, the authors performed clear and comprehensive experiments (such as pruning for potential application) and demonstration.

3. The method proposed in this paper is simple yet effective.

**Weaknesses:**

Apart from the given analysis in this paper, what interests me most is the different patterns between the supervised ViTs and self-supervised MAE, as demonstrated in `Fig.2` & `Tab.1`. At beginning, I personally attributed it to the lack of finetuning in self-supervised MAE, while the author pointed that this was also considered in `Line 218/328`. Although this almost **inverted distribution pattern** mentioned in `Sec 3.1's Preliminary Study` does not seem to have been resolved in `Tab.1`, the neuron path in MAE plays a much greater role than in supervised ViTs with lager *Accuracy Deviation*.

Perhaps this issue is beyond the scope of this paper, but this phenomenon worth a targeted analysis or discussion (maybe in future work).

**Questions:**

The formula in `Sec.3.2`'s JAS may be a little confusing.
1. The $w_{i_l}^{l}$ used in Formula(2)&(3) could be more simplified if the two $l$ are always same.
2. A set of neuron $w_{i_l}^{l}$, original values $\bar{w}\_{i_l}^{l}$, assigned value $\hat{w}_{i_l}^{l}$ could be more clearly defined before Formula(3).
3. JAS in Formula(2) is difficult to understand, pseudo code like `Algorithm 1` can be helpful.

I would be more than willing to reconsider my score based on the author's response to above `Weaknesses` & `Questions`.

---

> ### Author Response · Authors · 2024-11-20
>
> ### Weakness 1: Apart from the given analysis in this paper, what interests me most is the different patterns between the supervised ViTs and self-supervised MAE, as demonstrated in Fig.2 & Tab.1. At beginning, I personally attributed it to the lack of finetuning in self-supervised MAE, while the author pointed that this was also considered in Line 218/328. Although this almost inverted distribution pattern mentioned in Sec 3.1's Preliminary Study does not seem to have been resolved in Tab.1, the neuron path in MAE plays a much greater role than in supervised ViTs with lager Accuracy Deviation.
>
> 1. Thanks for pointing out this interesting experimental phenomena in our preliminary study! It demonstrates the inbalance caused by treating whole neurons equally.
> 2. The larger accuracy deviation of MAE may be caused by the different usage of tokens. In supervised ViT, the classification head will use the CLS token to do the final classification, while for the MAE, the classification head will utilize the global token, which is the mean of all the image tokens except the CLS token, which may lead to a better influence of the model performance after the modification.
> 3. But indeed, it's a interesting extension topic and we shall discuss more in the future works. Thanks again for your suggestion!

---

> ### Author Response · Authors · 2024-11-20
>
> ### Question 1: The formula in Sec.3.2's JAS may be a little confusing.
> ### 1. The $w^l_{i_l}$ used in Formula(2)&(3) could be more simplified if the two $l$ are always same.
> ### 2. A set of neuron $w^l_{i_l}$, original values $\overline{w}^l_{i_l}$, assigned value $\hat{w}^l_{i_l}$ could be more clearly defined before Formula(3).
> ### 3. JAS in Formula(2) is difficult to understand, pseudo code like Algorithm 1 can be helpful.
>
> 1. Thanks so much for your valuable questions and suggestions! Here is the formula of JAS,
> $$\text{JAS}(w _ {i _ 1}^1, w _ {i _ 2}^2, ..., w _ {i _ N}^N)= \sum _ {n=1}^N \overline{w} _ {i _ n}^n \int _ {\alpha=0}^{1} \sum _ {l=1}^N\frac{\partial F_ x(\alpha\overline{w} _ {i _ 1}^1, \alpha\overline{w} _ {i _ 2}^2, ..., \alpha\overline{w} _ {i _ N}^N)}{\partial w _ {i _ l}^l} d \alpha $$ For the first issue, yes, the upper and lower $l$ are always the same, the reason why we utilized such a reduntant notation is that in different layer, the selected neurons may not be the same. For example, $i_1$ isn't necessarily same as $i_2$, when $l$ equals to $1$ and $2$. So if we deleted the $l$ in the $i$, the formula may give rise to an ambiguity, since $w^1_i$ and $w^2_i$ share the same $i$.
> 2. For the second one, sorry for the ambiguity it may cause! Here is the detailed definition of the 3 variants. $w^l_{i_l}$ represents the $i_l$-th neuron located in $l$-th Transformer block's FFN module (or $l$-th layer in our paper notation). For example, $l=3$, and $i_l=i_3=47$, then $w^3_{47}$ means the $47$-th neuron in the FFN module of $3$-rd block of the given vision Transformer model. As for the original value $\overline{w}^l_{i_l}$, it represents the value of $w^l_{i_l}$ without modification. Note that in our definition, we say given an input $x$, which means the formula is input-related, the original value is tied with the given input. For example, for a pretrained vision Transformer, we feed in a cat image, and then we can grab the original value of neuron $w^3_{47}$, which is $\overline{w}^3_{47}$. Andfor the assigned value, since we need to calculate the integral of the function as joint integrated gradient, we will assign value to the selected neuron, which is $\hat{w}^l_{i_l}=\alpha \overline{w}^l_{i_l}$, controled by $\alpha \in [0, 1]$.
> 3. For the third question, we sincerely apologize for the trouble it may cause! As we mentioned in Def 1, for the sake of computational convinience, we use the Riemann approximation to estimate the continuous integral, which is $\widetilde{JAS}$. Thus, here is the pseudo code of the $\widetilde{JAS}$.
>     ```
>     Given the pretrained model F with L layers and the input x,
>     we want to calculate the JAS approximation of first N layers.
>     The selected neuron list n_l = [l_1, l_2,...,l_N], N <= L,
>     and the pre-fixed sampling step is m, we have the following
>     algorithm to calculate the JAS approximation:
>     JAS = 0, all neuron matrix w, w[l, idx] represent the
>     idx-th neuron at layer l, w_o[l, idx] represent the original
>     idx-th neuron at layer l
>     for k in range(m):
>         for l in range(N):
>             assign w[l, n_l[l]] with k/m * w_o[l, n_l[l]].
>         temp_sum = 0
>         for l in range(N):
>             temp_sum += d F(w)/ d w[l, n_l[l]] using gradient calculation.
>         JAS += temp_sum
>     for j in range(N):
>         JAS *= w_o[j, n_l[j]]
>     JAS /= m
>     ```
>     In the implementation, we form the $m$ step of integrated gradient as a whole batch and do gradient calculation.
> 4. We designed Eq. 2 for assessing the full contribution of the target neuron set, which we targeted at the individual neuron at each layer as "neuron path" in Definition 2 of our paper.
>
> ***
>
> In short, we sincerely thank you for providing insightful and valuable comments and suggestions. We are also open to further  discussion for improving the quality of our paper!

---

> > ### Comment · Reviewer_K2QS · 2024-11-25
> > **Response To Authors' Rebuttal.**
> >
> > Thanks for the authors' detailed rebuttal, which have addressed my previous concern.

---

> > > ### Author Response · Authors · 2024-11-25
> > >
> > > We are grateful for your acknowledgment of our work and for your valuable suggestions and perceptive observations. Should you have any further concerns or questions, we would be delighted to discuss them with you in order to enhance the quality of our paper.

---

### Official Review · Reviewer_BYaN · 2024-10-30

**Soundness:** 2
**Presentation:** 2
**Contribution:** 2
**Rating:** 6
**Confidence:** 4

**Summary:**

The paper introduces a method to uncover influential neurons across layers in Vision Transformer (ViT) models. It identifies paths of significant neurons from input to output, aiming to enhance understanding of how these models process information internally. The method, called "Neuron Path," employs a layer-progressive approach combined with a novel joint attribution score (JAS), designed to measure the collective influence of selected neurons on model predictions.
The paper proposes the Neuron Path method, which identifies key neuron paths that significantly contribute to the model’s decision-making process. This approach provides insights into information flow within the ViT model by tracing the most impactful neurons through each layer.

**Strengths:**

The Neuron Path method adopts a layer-progressive approach, which systematically identifies key neurons layer by layer. This method is a step toward understanding how information propagates through Vision Transformers (ViTs), aligning with the need for models that help illustrate the inner workings of transformers more clearly.
The joint attribution score provides a collective measure of neuron importance, enhancing typical neuron-based methods that often focus on single neurons. Although the score parallels existing joint attribution metrics, it offers a practical framework that reflects the relative importance of neurons throughout the model.

**Weaknesses:**

1. The paper introduces the "Neuron Path" method, which employs a layer-progressive approach to identify influential neurons across Vision Transformer (ViT) layers. However, this method significantly overlaps with established attribution techniques like [1， 2， 3]. The authors could clarify how their approach advances beyond these existing methods.

[1] HINT: Hierarchical Neuron Concept Explainer, CVPR, 2022.
[2] Neuron Shapley: Discovering the Responsible Neurons, NeurIPS, 2020.
[3] WWW: A Unified Framework for Explaining What Where and Why of Neural Networks by Interpretation of Neuron Concepts, CVPR, 2024.


2. While the paper underscores the structural complexity of ViTs as a barrier to interpretability, it narrowly investigates only one neuron type—the output of the first linear layer in each Transformer's FFN module. This limited focus raises questions about the motivation: effective explainability typically aims to enhance interpretability for either the model’s input-output relationship or its internal operations holistically. The choice to explore only specific layers restricts the scope, potentially overlooking valuable insights into other model layers or their contributions.

3. Relying solely on joint attribution scores to assess neuron influence, though innovative, lacks breadth. Incorporating additional metrics—such as concept coherence or task-specific performance indicators—could provide a more comprehensive view of the Neuron Path approach’s effectiveness. Moreover, the experimental focus on ViTs for image classification lacks exploration into generalizability. Extending this analysis to other models and tasks, like object detection, could demonstrate the broader utility of the neuron paths identified here.

4. The assumption that neurons in ViT FFNs are largely redundant warrants more rigorous testing. Studies show that FFN layers encode high-level visual features essential for robust model performance, and not all neurons in these layers may be extraneous. For example, Intriguing Properties of Vision Transformers (NeurIPS, 2021) and others illustrate that FFNs in ViTs play a significant role in preserving semantic information across layers, making them central to effective representation learning. Testing this redundancy hypothesis more thoroughly could yield valuable insights into FFN layer functionality.

**Questions:**

See weakness

---

> ### Author Response · Authors · 2024-11-20
>
> ### Weakness 1: The paper introduces the "Neuron Path" method, which employs a layer-progressive approach to identify influential neurons across Vision Transformer (ViT) layers. However, this method significantly overlaps with established attribution techniques like [1， 2， 3]. The authors could clarify how their approach advances beyond these existing methods.
> [1] HINT: Hierarchical Neuron Concept Explainer, CVPR, 2022. [2] Neuron Shapley: Discovering the Responsible Neurons, NeurIPS, 2020. [3] WWW: A Unified Framework for Explaining What Where and Why of Neural Networks by Interpretation of Neuron Concepts, CVPR, 2024.
>
> 1. The three papers you mentioned are good supplementary materials to our method. But they are quite different to our proposed methods.
> 2. For the first one, HINT method, it's a method that related feature map with the visual concepts， which can be view as an extension of feature map visualization. As mentioned in our paper, previous explainability works in vision models are mostly focus on visualization or saliency map. Objectively this does give a good description of the model's concerns, but it's relatively to quantify the effect of neurons on the model's reasoning.
> 3. As for the second paper, we already mentioned it in our related works. Neuron shapley is a computational heavy method, and it focused on the individual module part within the model. This is indeed a subtle approach, but the granularity of neurons is too coarse compared to ours and their optimization goals often focus on model output metrics, failing to fully elucidate the circulation of model knowledge and concepts.
> 4. And the last paper is the continuation of CLIP-Dissect [1], which is discussed in our related works. It externally provided powerful models, such as CLIP, rather than intrinsic model properties. And again, this methods failed to localize the neurons that are important to the model's inference, and faces the challenge to perform quantitative analytical experiments.
> 5. To summarize, our approach in still different from the papers you mentioned. Our approach, (i) focuses on the neurons, which is a more fine-grained neuron profiling rather than the whole module; (ii) tries to figure out the intrinsic mechanism of the model and the effect of information transfer within the model on the model's inference, rather than purely semantic focus; (iii) does not need the support of external models, and can be directly carried out to perform the quantitative analysis.
>
> [1] Oikarinen, Tuomas, and Tsui-Wei Weng. "Clip-dissect: Automatic description of neuron representations in deep vision networks." arXiv preprint arXiv:2204.10965 (2022).
>
> ### Weakness 2: While the paper underscores the structural complexity of ViTs as a barrier to interpretability, it narrowly investigates only one neuron type—the output of the first linear layer in each Transformer's FFN module. This limited focus raises questions about the motivation: effective explainability typically aims to enhance interpretability for either the model’s input-output relationship or its internal operations holistically. The choice to explore only specific layers restricts the scope, potentially overlooking valuable insights into other model layers or their contributions.
>
> 1. Indeed in our proposed method the neuron we investgated located in FFN layer, but it doesn't mean other layer is overlooked. FFN layers process the full information from the previous components. Quite the opposite, one of the reasons we study the FFN layer is that existing interpretability analysis work tends to focus only on the attention module or the convolution module, while neglecting the role of the FFN in supporting the network structure.
> 2. And in the previous paper[1], it pointed out that FFN module is also an essential memory part for the transformer structure. By exploring FFNs, we discovered many interesting properties, such as intraclass aggregation and interclass similarity, that have not been explored in other work exploring other modules.
> 3. For each Attention module, the final processing of the information is in the FFN, our approach does not look at each module in isolation, but explores the flow of information in the model as a whole, we do not miss the core flow of information in the attn, it is a complete exploratory approach, and since we study the whole model, our approach is likewise holistically of the study.
> 4. But again, your suggestion makes sense, and we'll further explore the other parts like Attention module in subsequent work, but this is orthogonal to what we're currently working on.
>
> [1] Geva, Mor, et al. "Transformer feed-forward layers are key-value memories." Proceedings of the 2021 Conference on Empirical Methods in Natural Language Processing (2021).

---

> > ### Comment · Reviewer_BYaN · 2024-11-26
> >
> > Thank you for your detailed response to my comment regarding the focus on neurons in the final FFN layer. While I understand your reasoning for prioritizing the FFN layer in your analysis, I believe your rebuttal does not fully address the concerns raised, and certain aspects of your argument remain unclear.
> >
> > 1. Your method claims to focus on understanding neuron importance within Vision Transformers (ViTs). However, by restricting the analysis to the final FFN layer, your study does not comprehensively address the broader dynamics of neuron importance across the entire model.
> >
> > 2. Your statement that exploring other components, such as attention modules, is orthogonal to your current work is contradictory to the broader claims made in your manuscript. If the goal of the study is to evaluate neuron importance, then attention modules—critical components of the Vision Transformer architecture—cannot be considered orthogonal.

---

> > > ### Author Response · Authors · 2024-12-01
> > >
> > > Thanks for your further comments!
> > >
> > > 1. In response to your initial inquiry, our methodology does not examine FFNs in isolation. Instead, it considers the flow of information within the model as a whole, through the neurons of the FFN. To some extent, the role of attention has been considered by our mechanistic explanation. The following is a more detailed explanation, by expanding our definition of Joint Attribution Score (JAS),
> > > $$\text{JAS}(w _ {i _ 1}^1, w _ {i _ 2}^2, ..., w _ {i _ N}^N) = \sum _ {n=1}^N \overline{w} _ {i _ n}^n \int _ {\alpha=0}^{1} \sum _ {l=1}^N\frac{\partial F _ x(\alpha\overline{w} _ {i _ 1}^1, \alpha\overline{w} _ {i _ 2}^2, ..., \alpha\overline{w} _ {i _ N}^N)}{\partial w _ {i _ l}^l} d \alpha .$$
> > > Focusing on the gradient part $\frac{\partial F _ x}{\partial w _ {i _ l}^l}$, we can define $z _ p$, which represents the intermediate variables (including those in attention layers) along the computational paths from $w _ {i _ l}^l$ to the model output. And we can generalize our fomulae into,
> > > $$\text{JAS}(w _ {i _ 1}^1, w _ {i _ 2}^2, ..., w _ {i _ N}^N) = \sum _ {n=1}^N \overline{w} _ {i _ n}^n \int _ {\alpha=0}^{1} \sum _ {l=1}^N \sum _ {z _ {p _ l} \in P _ l}\frac{\partial F _ x(\alpha\overline{w} _ {i _ 1}^1, \alpha\overline{w} _ {i _ 2}^2, ..., \alpha\overline{w} _ {i _ N}^N)}{\partial z _ {p _ l}} \frac{\partial z _ {p _ l}}{\partial w _ {i _ l}^l} d \alpha ,$$
> > > where $P _ l$ represents the set of all the intermediate variables along the computational paths from $w _ {i _ l}^l$ to the model output. In light of the fact that our formulation represents a path through the entire model, it follows that constant iterative operations will take into account every module in the model. This is in contrast to the approach of looking at FFNs in isolation, which is the approach of our baseline Activation method, or considering only downstream modules after the target layer, which is the approach of our preliminary study method Knowledge Neuron and the other baseline method Influence Pattern.

---

> > > ### Author Response · Authors · 2024-12-01
> > >
> > > 2. In response to your second inquiry, sorry for the confusion, the two approaches (i.e., studying FFN and attention module) are considered orthogonal because the roles of the FFN and attention modules within the model are somehow distinct, rather than because our methods are non-transferable. Our method can be applied to FFN neurons, as detailed in the paper, and it is also generalizable to attention modules. According to your suggestion, we have additionally conducted a supplementary experiment transferring our method to attention module, the results are listed as below. It has illustrated that our method is generalizable to other components in vision models, and it has a distinct advantage over benchmark methods in this experiment. Specifically, our method identifies the neurons within attention mechanism that are truly significant, as demonstrated in the subsequent experiments.
> > >
> > >     | Metrics                            | Methods            | Target Models |          |
> > >     |------------------------------------|--------------------|---------------|----------|
> > >     |                                    |                    | ViT-B-16      | ViT-B-32 |
> > >     | Joint Attribtution Score $\uparrow$           | Activation         | -0.0018       | -0.0069  |
> > >     |                                    | Neuron Path (ours) | **0.3287**       | **0.4983**   |
> > >     | Removal Accuracy Deviation (%)  $\downarrow$    | Activation         | -0.04         | 0        |
> > >     |                                    | Neuron Path (ours) | **-1.76**         | **-2.32**    |
> > >     | Enhancement Accuracy Deviation (%) $\uparrow$  | Activation         | 0.09          | 0.09     |
> > >     |                                    | Neuron Path (ours) | **1.47**         | **2.3**      |
> > > *Table 1: Comparison of methods on studying attention module in vision Transformer models. $\uparrow (\downarrow)$ means that the higher (lower), the better. The best results are in bold font.*
> > >
> > > **Experiment Setting**: Due to time constraints, we selected two models, ViT-B-16 and ViT-B-32, as our experimental subjects. We also chose attention output as the neuron selection. We follow the experiment setting the same as Section 4.2, Table 1, and apply Activation method as baseline method. It should be noted that this is merely a basic numerical analysis of the experiment. A deeper understanding of neuron behavior within the attention module requires exploring additional factors, such as the criteria for neuron selection, the roles and properties of individual neurons, and the distinctions between attention and FFN neurons. These avenues of investigation open up broader discussions about the inner workings of vision Transformers, particularly from the perspective of compound modules. We acknowledge these aspects and plan to address them as part of future work.
> > > The preceding explanation of orthogonal may have introduced some ambiguity; therefore, we have included the aforementioned experiment for clarification. Further investigation into the neuron study of attention is warranted to facilitate a more comprehensive discourse. We are grateful for your valuable suggestion. We will put these experiment results into the appendix in revision of our paper. It has not doubt that these experiments will enhance the analysis of our work. We will also refine the claim and description of our paper according to your suggestions.

---

> > > ### Author Response · Authors · 2024-12-01
> > >
> > > 3. Moreover, we would like to briefly reiterate the rationale behind our selection of FFN and its significance to the model. Recent research indicates that FFNs play a critical role in factual knowledge representation within transformer models [1, 2]. They demonstrate that FFN layers function as key-value memories, effectively storing associations between concepts (keys) and their corresponding information (values), mirroring the organization of factual knowledge in human cognition. Further, Meng et al. [3] conducted the experiments on different modules within a Transformer, finding that disrupting FFN layers significantly impairs factual recall, while disrupting self-attention layers has minimal impact. This suggests a direct role for FFNs in storing and accessing factual information. These findings are consistent with Meng et al. [4], who localized factual associations within FFN modules, corroborating the observations of Geva et al. [1, 2].
> > >
> > > We sincerely appreciate your recognition of our method's contribution to studying the joint effects of neurons within the inner workings of vision Transformers, as well as the novelty of our proposed joint attribution score as a collective measure of neuron importance. We are equally grateful for your invaluable suggestions and insightful observations, which we will carefully incorporate into our revised paper.
> > >
> > > [1] Geva M, Schuster R, Berant J, et al. Transformer Feed-Forward Layers Are Key-Value Memories[C]//Proceedings of the 2021 Conference on Empirical Methods in Natural Language Processing. 2021: 5484-5495.
> > > [2] Geva M, Caciularu A, Wang K, et al. Transformer Feed-Forward Layers Build Predictions by Promoting Concepts in the Vocabulary Space[C]//Proceedings of the 2022 Conference on Empirical Methods in Natural Language Processing. 2022: 30-45.
> > > [3] Meng K, Sharma A S, Andonian A J, et al. Mass-Editing Memory in a Transformer[C]//The Eleventh International Conference on Learning Representations. 2023.
> > > [4] Meng K, Bau D, Andonian A, et al. Locating and editing factual associations in GPT[J]. Advances in Neural Information Processing Systems, 2022, 35: 17359-17372.

---

> > > > ### Comment · Reviewer_BYaN · 2024-12-02
> > > >
> > > > Thanks for the authors' additional experiments and explanation, I have no further questions.

---

> > > > > ### Author Response · Authors · 2024-12-03
> > > > >
> > > > > We are grateful for your acknowledgment of our work and for your valuable suggestions. In order to make our series of works more complete and meaningful, we shall explore further the inner mechanisms of the attention module and other modules in future. We would be delighted to discuss any further concerns or questions you may have for the sake of enhancing the quality of our paper.

---

> ### Author Response · Authors · 2024-11-20
>
> ### Weakness 3: Relying solely on joint attribution scores to assess neuron influence, though innovative, lacks breadth. Incorporating additional metrics—such as concept coherence or task-specific performance indicators—could provide a more comprehensive view of the Neuron Path approach’s effectiveness. Moreover, the experimental focus on ViTs for image classification lacks exploration into generalizability. Extending this analysis to other models and tasks, like object detection, could demonstrate the broader utility of the neuron paths identified here.
>
> 1. First of all, thank you for recognizing the novelty of our method.
> 2. Our experiments show that, the found neuron paths, discovered by our joint attribution score, also reveal some concept coherence, such as the neuron paths illustrated clustering properties within the same semantic class (Finding 3 in Sec. 4.3), and the neuron paths share similarity of conceptually similar classes (Finding 4 in Sec. 4.4.). And the evaluation of accuracy deviation used in experiments of Sec. 4.2 is also task-specific.
> 3. Moreover, our work is more of a novel mechanistic interpretability methodology, which demonstrates the interesting findings of the important neuron at each layer. Our work explores the correlation between the features of neuron paths and the internal mechanisms of the model, with a focus on the methodology and the construction of a general framework. This approach can be extended to a larger scope, which is a very worthwhile direction to explore, but due to scope limitations, we are not able to cover all the tasks in this paper. We can extend our approach to other tasks because of its generality of the framework from the analysis perspective.
>
> ### Weakness 4: The assumption that neurons in ViT FFNs are largely redundant warrants more rigorous testing. Studies show that FFN layers encode high-level visual features essential for robust model performance, and not all neurons in these layers may be extraneous. For example, Intriguing Properties of Vision Transformers (NeurIPS, 2021) and others illustrate that FFNs in ViTs play a significant role in preserving semantic information across layers, making them central to effective representation learning. Testing this redundancy hypothesis more thoroughly could yield valuable insights into FFN layer functionality.
>
> 1. What we mean is that for a particular type of input, the FFN module is not utilized in its entirety, but only some of the neurons are involved in the transmission of information. That is, for the FFN module, the neurons are not equivalent in the transmission of semantic information, but are related to it. Admittedly, we're a bit off on the redundancy claim.
> 2. For a model that learns a multitude of visual information, different neurons in the FFN module encode different information, but only some of it are important for the very type of input, as reflected in our pruning experiments. In the previous paper [1], authors also found that the computational cost of FFN is expensive and claimed that the therein redundancy of FFN is usually overlooked. We want to convey that similar to intra-brain structures, where specific information is associated with specific pathways and through statistical neuron activation, visual models have the potential to pruning. In the language models, there is knowledge neurons [2] as well to analysis the sparse concept encoding, but in the different perspactive to ours.
> 3. It's an insightful suggestion and due to the limitation of paper scope, we can't start a related experiment. But in future related work, we will delve into this.
>
> [1] Xu, Haiyang, et al. "Vision transformer with attention map hallucination and ffn compaction." arXiv preprint arXiv:2306.10875 (2023).
> [2] Damai Dai, Li Dong, Yaru Hao, Zhifang Sui, Baobao Chang, and Furu Wei. 2022. Knowledge Neurons in Pretrained Transformers. In Proceedings of the 60th Annual Meeting of the Association for Computational Linguistics (Volume 1: Long Papers), pages 8493–8502, Dublin, Ireland. Association for Computational Linguistics.
>
> ***
>
> In short, we sincerely thank you for providing insightful and valuable comments and suggestions. We are also open to further  discussion for improving the quality of our paper!

---

### Official Review · Reviewer_7H8A · 2024-11-03

**Soundness:** 3
**Presentation:** 2
**Contribution:** 2
**Rating:** 6
**Confidence:** 3

**Summary:**

The paper introduces Neuron Path as a novel method for understanding Vision Transformers (ViT), focusing on the flow of information through the Feed-Forward Network (FFN) within the transformer encoder. Unlike methods that highlight input regions, Neuron Path examines influential neurons within the FFN across layers to reveal how information is processed. The authors propose the Joint Attribution Score (JAS), a gradient-based metric that measures the collective influence of neuron groups on the model’s output, along with a layer-by-layer selection method to identify impactful neurons, forming a path that traces critical information flow from input to output. Experiments show that Neuron Path effectively identifies neurons crucial for accurate predictions, with class-level analysis indicating that similar neurons activate for related classes, suggesting an internal organization of semantic information. Additionally, the authors find Neuron Path valuable for model compression, as retaining only essential neuron paths allows for nearly the same performance, revealing redundancy in Vision Transformers that could be leveraged for pruning and optimization.

**Strengths:**

1. The paper introduces the innovative concept of "Neuron Path," which identifies influential neuron paths specifically within the Feed-Forward Network (FFN) of vision transformers (ViTs). It also presents the Joint Attribution Score (JAS), a gradient-based metric that quantifies the collective influence of neuron groups across layers. Additionally, the Layer-Progressive Neuron Locating Algorithm is proposed to trace influential neuron paths efficiently.

2. The research methodology is robust, with comparisons against established baselines such as "Activation" and "Influence Pattern." The study includes comprehensive experiments, such as neuron intervention and model pruning, which provide strong empirical support for the effectiveness of the Neuron Path method.

3. The paper is well-structured and clearly explains its concepts, with well-defined terms, illustrative figures, and organized sections that enhance readability and understanding.

4. The study advances the interpretability of vision transformers by revealing insights into model behavior, potential for model compression, and implications for real-world applications. It fosters the development of more transparent and efficient vision models by uncovering redundancy that can be leveraged for pruning.

**Weaknesses:**

1. The current analysis is limited to neurons within the Feed-Forward Network (FFN) component of the Vision Transformer (ViT) encoder. Although FFNs are shown to play a critical role in encoding semantic concepts, excluding the self-attention mechanism from the Neuron Path analysis may lead to an incomplete understanding of information flow within the model, as it overlooks potential interactions between self-attention and FFN modules.

2. The study is focused solely on image classification, a discriminative task, leaving the applicability of Neuron Path to other vision tasks unexplored.

3. The paper does not explore theoretical connections between JAS and established interpretability frameworks, such as information theory or causal inference. Relating JAS to concepts like information bottlenecks or causal pathways could provide a stronger theoretical foundation, enhancing the rigor and interpretability of the findings.

4. While the paper suggests a potential link between neuron paths and semantic information, it does not examine this relationship in depth. Conducting a detailed analysis of how neuron paths correlate with human-interpretable concepts could significantly enhance the model’s interpretability.

5. The paper primarily focuses on identifying neuron paths and their impact on model performance but does not delve into the properties of these paths. Investigating the stability of neuron paths across varying inputs or their evolution during training could provide valuable insights into the model's robustness and behavior. However, due to the computational expense mentioned by the authors, such analyses may be challenging to conduct.

**Questions:**

1. The proposed method focuses specifically on ViTs and is limited to FFNs. Could this approach be generalized to other model architectures or extended to vision tasks beyond image classification?

2. The authors assert that JAS computation is computationally expensive. Are there possible optimizations or approximations that could be investigated to reduce the computational load and enhance the scalability of the Neuron Path method for larger vision transformer models? Furthermore, why not use zero-order derivatives as a substitute for partial derivatives in calculating JAS, as I believe they could achieve a comparable effect?

3. The paper introduces JAS to measure joint neuron influence, yet a deeper theoretical understanding is needed. Could the authors clarify how maximizing JAS identifies crucial information pathways? Additionally, could they link JAS to information theory concepts like mutual information or information bottleneck to justify its use in measuring influential pathways?

---

> ### Author Response · Authors · 2024-11-20
>
> ### Weakness 1: The current analysis is limited to neurons within the Feed-Forward Network (FFN) component of the Vision Transformer (ViT) encoder. Although FFNs are shown to play a critical role in encoding semantic concepts, excluding the self-attention mechanism from the Neuron Path analysis may lead to an incomplete understanding of information flow within the model, as it overlooks potential interactions between self-attention and FFN modules.
>
> 1. In comparison to other non-FFN interpretable work on visual models, our approach is more focused on the intrinsic mechanisms of the model and information transfer. In contrast, other studies tend to employ visualization techniques, such as heat maps, to explore the characteristics of the module of interest. While the visualization-based approach is relatively simple and straightforward, it lacks the capacity for in-depth exploration of the intrinsic mechanisms of the model. Furthermore, it is difficult to conduct direct evaluation without relying on external datasets or models.
> 2. To the best of our knowledge, our approach is the first to address this gap by exploring the neuronal connections within FFNs to study the intrinsic information transmission of visual models. The exploration yielded several noteworthy findings pertaining to neuron paths, including intraclass aggregation and interclass similarity. These observations align with the intuitive understanding of the human visual system, thereby reinforcing the importance of our study.
> 3. In a previous paper [1], the authors highlighted the importance of the FFN module in the Transformer structure as a crucial component of memory. Accordingly, this paper focuses on the FFN module due to the scope limitations imposed by the aforementioned considerations.
> 4. It's a really good comment, and in the future works we will explore more about the neurons within other modules of Transformer model, such attention mechanism. We believe that our proposed method also applies to other structures due to its generality.
>
> [1] Geva, Mor, et al. "Transformer feed-forward layers are key-value memories." Proceedings of the 2021 Conference on Empirical Methods in Natural Language Processing (2020).
>
> ### Weakness 2: The study is focused solely on image classification, a discriminative task, leaving the applicability of Neuron Path to other vision tasks unexplored.
>
> 1. In addition to a brief discussion of the classification task, we also applied neuron paths to an attempt at the pruning task, which proved successful.
> 2. This article presents the concept of neuron paths and the methodology for exploring them. The focus is on examining the mechanisms of neuron paths and analyzing the properties of the model, including intraclass aggregation, interclass similarity, and potential applications such as pruning. The objective of this study is to present a general framework for neuron analysis that can be applied to a variety of tasks and models.
> 3. Due to space limitations and the focus of our research, we are unable to discuss comprehensive downstream tasks in this paper. However, we value your suggestions and will consider them in our future work. In future work, we intend to investigate additional downstream tasks, such as segmentation and generation.

---

> ### Author Response · Authors · 2024-11-20
>
> ### Weakness 3: The paper does not explore theoretical connections between JAS and established interpretability frameworks, such as information theory or causal inference. Relating JAS to concepts like information bottlenecks or causal pathways could provide a stronger theoretical foundation, enhancing the rigor and interpretability of the findings.
>
> 1. It's a really good comment. The intuition of our proposed method is visual pathway from neuronscience [1], which is a complex system of neurons that carries visual information from the retina to the brain. Moreover, the differences between our approach and the method you mentioned are from several different aspects.
> 2. The intuition behind the methods are different. We believe that in a visual model, a similar neuron path should exist from input to output to undertake the transmission and processing of visual information. This is slightly different from information bottleneck or causal pathway, which are based on the perspective of information theory and probability inference.
> 3. The optimization goal is different. For our approach, we are trying to find the neuron paths *within the model* that are most important for information transfer. It is structural that are the individual neurons at each model layer. While information bottleneck or causal pathway do not guarantee this hypothesis. And unlike some explainability works relying on information bottleneck need to minimize the target information component like representation [2], our approach is a parameter-free method and only need the target model itself for the probing. And information bottleneck are also used in visualization [3], which is also a different perspective of expainability to our approach.
> 4. There are still some similarities between these methods. For example, their explanatory goal can converge to find some key parts of the model. The difference is that, for example information bottleneck, looks for the smallest compressible subset the input to restore the information, while our approach focuses on the inner mechanism of the model.
>
> It's a insightfull comment, bridging these methods also worth exploration in the future.
>
> [1] Gupta, Mohit, Ashley C. Ireland, and Bruno Bordoni. "Neuroanatomy, visual pathway." (2020).
> [2] Wu, Tailin, et al. "Graph information bottleneck." Advances in Neural Information Processing Systems 33 (2020): 20437-20448.
> [3] Wang, Ying, Tim GJ Rudner, and Andrew G. Wilson. "Visual explanations of image-text representations via multi-modal information bottleneck attribution." Advances in Neural Information Processing Systems 36 (2023): 16009-16027.
>
> ### Weakness 4: While the paper suggests a potential link between neuron paths and semantic information, it does not examine this relationship in depth. Conducting a detailed analysis of how neuron paths correlate with human-interpretable concepts could significantly enhance the model’s interpretability.
>
> 1. It is really a good comment. In fact, since neuron paths are discovered based on some input images, which contain some semantic information with the ground-truth class label. The labels are human-interpretable and have already been leveraged in the analysis in Sec. 4.3, revealing the clustering properties of neuron paths within the same class (Finding 3) and high semantic similarity between that of different classes (Finding 4). From the results of the analysis, this semantic correlation exists to some extent, which is in line with our knowledge that neuron paths are channels that carry visual information transmission.
> 2. This reflects the fact that there is conceptual partitioning (path aggregation) within ViT, and that there are semantic associations of neurons within the model, explaining that there are specific regions of the neural network that are good at processing some specific semantic information. Moreover, the neuron semantic gap is significant for semantic information with large conceptual distinctions, while the semantic gap is small for fine-grained conceptual distinctions.
> 3. This is an interesting comment and raises our curiosity as to whether the model can leverage more human-interpretable concepts related to the discovered neuron path, such as automatically expanding the annotating on the input images as well as the corresponding neuron paths to gain deeper insights. We will further explore this direction.

---

> ### Author Response · Authors · 2024-11-20
>
> ### Weakness 5: The paper primarily focuses on identifying neuron paths and their impact on model performance but does not delve into the properties of these paths. Investigating the stability of neuron paths across varying inputs or their evolution during training could provide valuable insights into the model's robustness and behavior. However, due to the computational expense mentioned by the authors, such analyses may be challenging to conduct.
>
> 1. That's a really insightful comment! Indeed in the course of our experiments, it occurred to us whether the process of model training could be explored through the observation of neuron paths.
> 2. The issue is indeed a fascinating and significant one, but the current method's computational complexity and the scope of this paper prevent us from developing this research in this paper for the time being. This paper's focus is on proposing a general explainability framework and a property related to the neuron path. Nevertheless, future work will aim to develop more sophisticated methodologies to conduct this experiment.
>
> ### Question 1: The proposed method focuses specifically on ViTs and is limited to FFNs. Could this approach be generalized to other model architectures or extended to vision tasks beyond image classification?
>
> Please refer to the response to W2. We will discuss the utility of neuron mechanism in other model structure or tasks in the future work.
>
> ### Question 2: The authors assert that JAS computation is computationally expensive. Are there possible optimizations or approximations that could be investigated to reduce the computational load and enhance the scalability of the Neuron Path method for larger vision transformer models? Furthermore, why not use zero-order derivatives as a substitute for partial derivatives in calculating JAS, as I believe they could achieve a comparable effect?
>
> 1. It's a really good comment. This problem can be approached from two perspectives: the optimization of the implementation itself and the question of whether the zero-order derivative can be employed in its place, that is, why the first-order derivative is used.
> 2. In the specific implementation of the code, we have optimized it with a parallel computing method, including, but not limited to, saving the weights in advance, a parallel sampling step, and so on. A detailed account of the implementation process will be made available upon the code's public release.
> 3. In response to the second inquiry, we must conclude that the answer is negative. From an implementation perspective, zero-order derivatives are analogous to a weighted activation averaging method. In our benchmark, the activation method that relies solely on neuron activation information is not representative of neuron paths. The theoretical justification for the use of first-order derivatives for cumulative gradients is derived from the preceding paper [1]. In this paper, the authors provide a comprehensive rationale for why the integrated gradient is a superior form of attribution compared to the gradient. In conclusion, integrated gradients demonstrate sensitivity to the input and are independent of the model structure.
> 4. In regard to our methodology, as illustrated in Eq. 2, we extend the integrated gradient from a focus on a single neuron to encompass an entire neuron path. This allows for the representation of joint attribution and the measurement of the joint effect of a neuron path on the overall performance of the model.
>
> [1] Sundararajan, Mukund, Ankur Taly, and Qiqi Yan. "Axiomatic attribution for deep networks." International conference on machine learning. PMLR, 2017.
>
> ### Question 3: The paper introduces JAS to measure joint neuron influence, yet a deeper theoretical understanding is needed. Could the authors clarify how maximizing JAS identifies crucial information pathways? Additionally, could they link JAS to information theory concepts like mutual information or information bottleneck to justify its use in measuring influential pathways?
>
> 1. In our response to question 2, we presented the importance of the cumulative gradient for attribution. In order to maximize JAS, we employ the Layer-progressive Neuron Locating Algorithm, which reduces the complexity of the neuron search space by utilizing a greedy-like search algorithm to identify the locally optimal neurons in a layer-by-layer manner, and subsequently combines them to form the final neuron path that maximizes JAS.
> 2. With regard to the relationship between our proposed method and information theory, we have provided an explanation in weakness 3. This is a perceptive observation, and we intend to investigate it further in future research.
>
> ***
>
> In short, we sincerely thank you for providing insightful and valuable comments and suggestions. We are also open to further  discussion for improving the quality of our paper!

---

> > ### Comment · Reviewer_7H8A · 2024-11-24
> >
> > Thank you for the detailed response. Most of my concerns have been addressed. I look forward to seeing the authors extend the analysis to include self-attention mechanisms and Transformer modules, which would enhance the understanding of internal information flow and solidify the neuron path analysis framework. I think applying this framework to more challenging vision tasks could validate its generalizability and expand its practical utility. Overall, I will raise my rating to 6.

---

> > > ### Author Response · Authors · 2024-11-24
> > >
> > > We are grateful for your acknowledgment of the contribution of our paper and for raising the score. In light of your insightful feedback, we will continue to refine and enhance our approach in future endeavors, including but not limited to:
> > > 1. Broadening the scope of our methodology to encompass a broader range of vision tasks, such as segmentation task and generating task.
> > > 2. Extending our method to other components of vision Transformer models, investigating the performance and characteristics of neurons across diverse modules, such as Attention, to gain deeper insights into the intrinsic mechanisms of the model.
> > >
> > > We extend our gratitude once more for acknowledging our work and for raising our score. We also express our appreciation for your invaluable suggestions and perspicacious observations.

---

### Official Review · Reviewer_jrVR · 2024-11-04

**Soundness:** 3
**Presentation:** 3
**Contribution:** 2
**Rating:** 6
**Confidence:** 3

**Summary:**

This paper addresses identifying the neuron path within Vision Transformers (ViT) that comprises the most impactful neurons. Two methods are utilized to identify a neuron pathway. The proposed joint attribution score assesses the influence of the neuron pathway on the network's inference process. Another method referred to as layer-progressive neuron locating identifies the most significant neuron pathway for predicting the target image. The proposed method demonstrates its effectiveness in identifying neuron pathways. This paper presents a range of applications that leverage the proposed method.

Thank you for the authors' response. I think they answered my question adequately. Thus, I keep my score as a weak accept.

**Strengths:**

This paper presents innovative concepts for identifying influential neuron pathways. The proposed approach is capable of identifying the trajectory of neurons by analyzing the gradient magnitude. This method has the potential for application across diverse architectural frameworks.

This paper demonstrates the applicability of the proposed method across a range of scenarios. Initially, Figure 4 visualizes the pathways of neurons categorized by class. The visualization validates that various classes utilize unique neural pathways. Subsequently, Figure 5 demonstrates the class relationships derived from the discovered neuron pathways. This approach has the potential to facilitate the assessment of similarity between classes in forthcoming research.

**Weaknesses:**

This paper does not adequately present a thorough review of the existing literature regarding the proposed method. The reviewer, upon examining the current manuscript, encounters difficulty in clarifying the significance of the proposed method in relation to recent studies. The discussion should not merely enumerate related works; it should also engage in a comparative analysis of the proposed method to facilitate a clear understanding of its significance for the readers.

The manuscript requires enhancements in coherence for better clarity and flow. The abstract states, “showcasing that the found neuron paths have already preserved the model capability on downstream tasks.” However, there is a lack of research focused on downstream tasks.

The absence of crucial ablation studies for the proposed joint attribution score is a significant oversight in this paper. The proposed joint attribution score represents a sophisticated approach that necessitates multiple iterations of back-propagation. To establish its efficacy, it is essential to conduct a comparison with a more straightforward baseline method and demonstrate its superiority over that baseline. Nonetheless, Table 1 presents a simultaneous variation of search methods and scores, which complicates the identification of the specific component contributing to performance improvement. An appropriate ablation study is essential to address the increase in search complexity associated with the proposed method.

This paper does not include comparisons with the baseline method in Figures 5 and 6. The analysis studies are commendable; however, it is essential that they also incorporate quantitative metrics in relation to the baseline for a more comprehensive evaluation. The sole quantitative metric presented in this paper is presented in Table 1. The author needs to incorporate more extensive assessments to demonstrate the advantages of the proposed method.

**Questions:**

How can we calculate the output by leveraging the neurons present in the initial N layers? Should we share the final classifier to compute the probability p of Eq 1?

It would be better to provide the meaning of Eq 2. Currently, there is no description for Eq 2. Also, the reviewer wonders about the rationale behind employing the sum of gradients as the proposed significant score.

---

> ### Author Response · Authors · 2024-11-20
>
> ### Weakness 1: This paper does not adequately present a thorough review of the existing literature regarding the proposed method. The reviewer, upon examining the current manuscript, encounters difficulty in clarifying the significance of the proposed method in relation to recent studies. The discussion should not merely enumerate related works; it should also engage in a comparative analysis of the proposed method to facilitate a clear understanding of its significance for the readers.
>
> Thank you for pointing this out. We have significantly refined the corresponding part about related work in the paper (**we marked the main changes in the updated manuscript in blue color**). And we briefly summarize that and position our work as below.
> 1. One category is that of visualization-based methods. One limitation of these approaches is the difficulty of conducting a quantitative evaluation. These methods are based on feature visualization, which is used to illustrate the pattern attention of the module. This represents a distinct perspective on explainability, which differs from our approach.
> 2. Another stream of research is neuron-based methods. The majority of previous studies have not considered the joint effect of a set of neurons, instead treating all neurons equally and independently. Moreover, the majority of research pertaining to the neuron-based approach is situated within the domain of natural language processing (NLP).
> 3. In conclusion, our method shifts the focus towards the collective contribution of selected neurons across different layers, thereby offering a more comprehensive understanding of model behavior and enhancing its explainability. Furthermore, it provides valuable insights into the model's decision-making process.
>
> ### Weakness 2: The manuscript requires enhancements in coherence for better clarity and flow. The abstract states, “showcasing that the found neuron paths have already preserved the model capability on downstream tasks.” However, there is a lack of research focused on downstream tasks.
>
> Thanks for the suggestion!
> 1. In this paper, we focus on the discovery and properties of neuron paths, including (i) the development of new concepts and algorithms, (ii) the discovery of properties such as intraclass aggregation and interclass similarity of the found neuron paths, as well as potential applications such as pruning, with the classification task being the core case study. But due to space constraints, we have not allocate enough space for more details of the downstream tasks. We'll further enhance our presentation for clearer claim.
> 2. In this paper we have discussed the two tasks of pruning and image classification, but you have made a valuable suggestion that in subsequent work we will further explore the nature and findings of neuron paths in wider applications in downstream tasks.
>
> ### Weakness 3: The absence of crucial ablation studies for the proposed joint attribution score is a significant oversight in this paper. The proposed joint attribution score represents a sophisticated approach that necessitates multiple iterations of back-propagation. To establish its efficacy, it is essential to conduct a comparison with a more straightforward baseline method and demonstrate its superiority over that baseline. Nonetheless, Table 1 presents a simultaneous variation of search methods and scores, which complicates the identification of the specific component contributing to performance improvement. An appropriate ablation study is essential to address the increase in search complexity associated with the proposed method.
>
> Thanks for your suggestion. Our existing experimental setup has been designed to accommodate ablation experiments.
> 1. The proposed method can be divided into two principal components: the number of neurons and the joint attribution score. Figure 2 in the preliminary study presents an ablation of the number of neurons, which reflects the imbalance in the distribution of single neuron attributions.
> 2. Table 1 in the Quantitative Comparison presents the results of ablation comparisons of attribution scores using two simple benchmark methods: activation and influence pattern. These comparisons demonstrate the efficacy of our approach in ablating JAS. This is not a simultaneous alteration of the search methodology and the metrics employed. Rather, it is a comparison of disparate methods on distinct models, with the three metrics representing three distinct comparison experiments.
> 3. In light of the fact that we are proposing a novel concept rather than a modification of the model structure, it is challenging to conduct further ablation of the methods or metrics.

---

> ### Author Response · Authors · 2024-11-20
>
> ### Weakness 4: This paper does not include comparisons with the baseline method in Figures 5 and 6. The analysis studies are commendable; however, it is essential that they also incorporate quantitative metrics in relation to the baseline for a more comprehensive evaluation. The sole quantitative metric presented in this paper is presented in Table 1. The author needs to incorporate more extensive assessments to demonstrate the advantages of the proposed method.
>
> 1. In our analysis, Figure 5 corresponds to class level semantic analysis trying to validate if the discovered neuron paths reveal some specific semantic similarity regarding the samples in semantically similar classes. This finding is novel and associates the discovered neuron paths with semantic concepts, which has not been explored in prior work on neural network mechanistic interpretability. The previous work has tended to directly assign concepts or knowledge to some individual neurons. It is not straightforward to directly compare different methods in this anlaysis. And we will keep thinking of that since this a promising exploration and may enhance our analysis.
> 2. As for Figure 6, we also conduct additional comparison using activation values as neurons, employing the identical configuration as that utilized in Section 4.2, wherein the topK activation values in each FFN block was selected as neurons. Concurrently, we conducted ablation experiments with path lengths to investigate the impact of varying neuron path lengths on the pruning effect. The depth denotes the depth of probing layers (neuron path length), and the mask represents the percentage of deleted neurons, excluding the selected neurons.
>
> | `depth` | `mask` |   Activation topK=5  | JAS topK=5 |
> |---------|--------|----------------------|------------|
> |       6 |    100 | 0.517991  | **0.521252**  |
> |       9 |    100 | 0.518137  | **0.527151**  |
> |      12 |    100 | 0.531976  | ***0.540702***  |
>
> The results demonstrate that our method outperforms the activation baseline, and that the depth of the neuron path is a also significant.
>
> [1] Chavan, Arnav, et al. "Vision transformer slimming: Multi-dimension searching in continuous optimization space." Proceedings of the IEEE/CVF Conference on Computer Vision and Pattern Recognition. 2022.
>
> ### Question 1: How can we calculate the output by leveraging the neurons present in the initial N layers? Should we share the final classifier to compute the probability p of Eq 1?
>
> 1. If understood correctly, your question is how we calculate the JAS for the first $N$ layers. The model structure is fixed, what we modified is the inner neurons, which leads to the change of output performance. Combining the definitions of Eq. 2,
> $$\text{JAS}(w _ {i _ 1}^1, w _ {i _ 2}^2, ..., w _ {i _ N}^N)= \sum _ {n=1}^N \overline{w} _ {i _ n}^n \int _ {\alpha=0}^{1} \sum _ {l=1}^N \frac{\partial F _ x(\alpha\overline{w} _ {i _ 1}^1, \alpha\overline{w} _ {i _ 2}^2, ..., \alpha\overline{w} _ {i _ N}^N)}{\partial w _ {i _ l}^l} d \alpha $$
> $F$ is a complete model structure, and after determining the first $N$ layers as parameters, our changes will only be made in the first $N$ layers, and the model structure of the latter is fixed.
> 2. So for the calculation of the JAS of the first $N$ layers, we will fix the rest $L-N$ blocks, just modify the first $N$ layers' neurons. And for the final classification head, it is shared across all the calculation process to compute the probability $p$ as well, thus ensuring the complete backpropagation of the gradient.

---

> ### Author Response · Authors · 2024-11-20
>
> ### Question 2: It would be better to provide the meaning of Eq 2. Currently, there is no description for Eq 2. Also, the reviewer wonders about the rationale behind employing the sum of gradients as the proposed significant score.
>
> 1. In Eq. 2, we extend and calculate integrated gradient, the cumulative gradient of the measure of interest (classification output) $F_x$ to the path of the target component (the neuron set $[\bar{w} _ {i _ 1}^1, \ldots, \bar{w} _ {i _ 1}^i, \ldots, \bar{w} _ {N _ 1}^N]$) ranging from zero effect ($\alpha=0$) to full effect ($\alpha=1$), given the model input $x$. This assesses the full contribution of the target neuron set, which we targeted at the individual neuron at each layer as "neuron path" in Definition 2 of our paper.
> 2. Refer to the previous work [1], the authors gave detailed reasons why the integrated gradient is a better form of attribution compared to the gradient. To summurize, integrated gradients satisfies the sensitivity to the input, vanilla gradient calculation may fail due to the activation function, since the change of input may not change the activation output. And it's invarient to the model strcture, while other methods like LRP [2] and DeepLift [3] given as examples in the paper failed due to the replacement of gradient.
> 3. And our proposed approach further extend the integrated gradient based attibution score, which enables it to detect the joint effect of a selected neruon path.
> 4. Thank you for your point. We will further enhance our presentation in the revised version of our paper.
>
> [1] Sundararajan, Mukund, Ankur Taly, and Qiqi Yan. "Axiomatic attribution for deep networks." International conference on machine learning. PMLR, 2017.
> [2] Binder, Alexander, et al. "Layer-wise relevance propagation for neural networks with local renormalization layers." Artificial Neural Networks and Machine Learning–ICANN 2016: 25th International Conference on Artificial Neural Networks, Barcelona, Spain, September 6-9, 2016, Proceedings, Part II 25. Springer International Publishing, 2016.
> [3] Li, Junbing, et al. "Deep-LIFT: Deep label-specific feature learning for image annotation." IEEE transactions on Cybernetics 52.8 (2021): 7732-7741.
>
> ***
>
> In short, we sincerely thank you for providing insightful and valuable comments and suggestions. We are also open to further  discussion for improving the quality of our paper!

---

### Official Review · Reviewer_TbDp · 2024-11-08

**Soundness:** 3
**Presentation:** 3
**Contribution:** 3
**Rating:** 6
**Confidence:** 3

**Summary:**

This study focuses on uncovering the inner workings of vision Transformer models by examining critical neurons that influence model inference. They introduce the Neuron Path approach, which uses a new joint neuron attribution measure to identify key neurons in information flow across layers. The analytical experiments highlight the significance of these neuron paths for model inference and provide insights into the internal operations of vision Transformers, enhancing the field's understanding of visual information processing.

**Strengths:**

This paper is relatively well-motivated as understanding vision transformer is a crucial issue. I also find the evaluations thorough. The strengths are as follows:

1,The target issues of the paper are meaningful and worth exploring.
2. The motivation is clear.
3.The paper is easy to follow.

**Weaknesses:**

1. In section 4.4, neuron path method is used in model compression. It would be better if this method is compared to the the pruning method.
2. This method aims at vision transformer. Can this method be utilized in NLP tasks?

**Questions:**

See Weaknesses

---

> ### Author Response · Authors · 2024-11-20
>
> ### Weakness 1: In section 4.4, neuron path method is used in model compression. It would be better if this method is compared to the the pruning method.
> 1. That's a good suggestion! We considered whether to include existing pruning methods as a comparison when designing our experiments, but to be fair, we did an exploratory experiment using only our method. We tried using ViT-Slim [1] as our compression baseline, here are the results.
>     | Method    | Acc       |
>     |:---------:|:---------:|
>     | ViT-Slim  | 9.96%     |
>     | JAS-Prune | 67.7%     |
>
>     For the sake of fairness, we let ViT-Slim focus only on FFNs as our approach. But it focuses more on the module weights, while our method focuses on the neuron itself. And for the dataset setup, we use the validation set split that is aligned with the one in our paper. ViT-Slim may be more efficient, but training on a limited dataset limits its performance. This is also a side effect of the fact that our method selects key neurons.
> 2. Furthermore, we also conduct additional comparison using activation values as neurons, employing the identical configuration as that utilized in Section 4.2, wherein the topK activation values in each FFN block was selected as neurons. Concurrently, we conducted ablation experiments with path lengths to investigate the impact of varying neuron path lengths on the pruning effect. The depth denotes the depth of probing layers (neuron path length), and the mask represents the percentage of deleted neurons, excluding the selected neurons.
>
>     | `depth` | `mask` |   Activation topK=5  | JAS topK=5 |
>     |---------|--------|----------------------|------------|
>     |       6 |    100 | 0.517991  | **0.521252**  |
>     |       9 |    100 | 0.518137  | **0.527151**  |
>     |      12 |    100 | 0.531976  | ***0.540702***  |
>
> The results demonstrate that our method outperforms the activation baseline, and that the depth of the neuron path is a also significant.
>
> [1] Chavan, Arnav, et al. "Vision transformer slimming: Multi-dimension searching in continuous optimization space." Proceedings of the IEEE/CVF Conference on Computer Vision and Pattern Recognition. 2022.
>
>
> ### Weakness 2: This method aims at vision transformer. Can this method be utilized in NLP tasks?
>
> 1. It's a valuable suggestion. In the field of NLP, attribution methods represent a prevalent approach to explainability. However, to the best of our knowledge, there are few neuronal interpretation methods in the NLP field that consider the joint attribution of neuron paths that span across model layers. This is one of the novel aspects of our approach.
> 2. Additionally, there is a conceptual similarity between images and words. A previous study [1] has explored the embedding similarity of the same concepts, thereby providing support for the generalization of our proposed method to the NLP domain. Additionally, the Transformer is a common structure utilized in natural language processing (NLP) as well.
> 3. In this paper, we focus on neuron paths in visual models. Due to space and time constraints, we do not explore the application of this new method in the NLP domain. However, we believe that this method has the potential to be applied to NLP tasks. For example, one might consider the connection between pre- and post-text, or the process of thinking and generating within language models. This will be a further topic of investigation in our future work.
>
> [1] Huh, M., Cheung, B., Wang, T. &amp; Isola, P.. (2024). Position: The Platonic Representation Hypothesis. Proceedings of the 41st International Conference on Machine Learning, in Proceedings of Machine Learning Research 235:20617-20642
>
> ***
>
> In short, we sincerely thank you for providing insightful and valuable comments and suggestions. We are also open to further  discussion for improving the quality of our paper!

---

### Meta-Review · Area_Chair_sTky · 2024-12-24

**Metareview:**

This work explores the importance of influential neuron paths in vision Transformers (ViTs). To achieve this, the authors propose a joint influence measure to evaluate the contribution of specific neuron sets to model outcomes, along with a layer-progressive neuron locating approach that efficiently identifies the most influential neurons at each layer of the target model. The reviewers unanimously acknowledge the innovation of introducing Neuron Paths as a means to enhance the understanding of ViTs, as well as the effectiveness of the proposed method. The paper is recognized for its clear writing, well-structured presentation, and comprehensive evaluation results. Most of the reviewers' concerns were thoroughly addressed, including requests for additional comparisons with other pruning methods, further explanation of the proposed JAS scores with pseudocode, and clarification of the paper’s contributions and distinctions from related works. Finally, all reviewers provided positive ratings, resulting in an average score of 6. Consequently, we decide to accept the paper. In addition, the authors are encouraged to incorporate the remaining reviewers' suggestions into the final version if time permits.

**Additional Comments On Reviewer Discussion:**

The authors provided detailed responses to each concern raised by the reviewers, addressing requests for additional comparisons with other pruning methods and ablation studies (TbDp, jrVR), explanations of the proposed JAS scores with accompanying pseudocode (jrVR, BYaN, K2QS), clarification of the paper’s contributions and distinctions from similar works (BYaN), and the application of the method to more tasks or ViT modules beyond FFN (7H8A). These thorough responses successfully led to positive ratings from all reviewers.

---

### Decision · Program_Chairs · 2025-01-22

Accept (Poster)